# Local projections of layer Vb-to-Va are more prominent in lateral than in medial entorhinal cortex

**Shinya Ohara[1,2]\*, Stefan Blankvoort[1], Rajeevkumar Raveendran Nair[1], Maximiliano J Nigro[1], Eirik S Nilssen[1], Clifford Kentros[1], Menno P Witter[1]\***

[1]Kavli institute for Systems Neuroscience, Center for Computational Neuroscience, Egil and Pauline Braathen and Fred Kavli Center for Cortical Microcircuits, NTNU Norwegian University of Science and Technology, Trondheim, Norway; [2]Laboratory of Systems Neuroscience, Tohoku University Graduate School of Life Sciences, Tohoku, Japan

**Abstract** The entorhinal cortex, in particular neurons in layer V, allegedly mediate transfer of information from the hippocampus to the neocortex, underlying long-term memory. Recently, this circuit has been shown to comprise a hippocampal output recipient layer Vb and a cortical projecting layer Va. With the use of in vitro electrophysiology in transgenic mice specific for layer Vb, we assessed the presence of the thus necessary connection from layer Vb-to-Va in the functionally distinct medial (MEC) and lateral (LEC) subdivisions; MEC, particularly its dorsal part, processes allocentric spatial information, whereas the corresponding part of LEC processes information representing elements of episodes. Using identical experimental approaches, we show that connections from layer Vb-to-Va neurons are stronger in dorsal LEC compared with dorsal MEC, suggesting different operating principles in these two regions. Although further in vivo experiments are needed, our findings imply a potential difference in how LEC and MEC mediate episodic systems consolidation.

**\*For correspondence:**
shinyaohara@gmail.com (SO);
menno.witter@ntnu.no (MPW)

**Competing interests:** The authors declare that no competing interests exist.

## Introduction

Everyday memories, which include information of place, time, and content of episodes, gradually mature from an initially labile state to a more stable and long-lasting state. This memory maturation process, called memory consolidation, involves gradual reorganization of interconnected brain regions: memories that are initially depending on hippocampus become increasingly dependent on cortical networks over time (*Frankland and Bontempi, 2005*). Although various models have been hypothesized for this systems-level consolidation, such as the standard consolidation model and multiple trace theory (*Nadel and Moscovitch, 1997*; *Squire and Alvarez, 1995*), they all share a canonical hippocampal-cortical output circuit via the entorhinal cortex (EC), which is crucial to mediate long-term memory storage and recall (*Buzsáki, 1996*; *Eichenbaum et al., 2012*). The existence of this circuit was originally proposed based on the ground-breaking report of a non-fornical hippocampal-cortical output route mediated by layer V (LV) of the EC in monkeys (*Rosene and Van Hoesen, 1977*), which was later confirmed also in rodents (*Köhler, 1985*; *Kosel et al., 1982*).

The EC is composed of two functionally distinct subdivisions, the lateral and medial EC (LEC and MEC, respectively). MEC processes allocentric, mainly spatial information, whereas LEC represents the time and content of episodes (*Deshmukh and Knierim, 2011*; *Hafting et al., 2005*; *Montchal et al., 2019*; *Tsao et al., 2018*; *Tsao et al., 2013*; *Xu and Wilson, 2012*). Despite these evident functional differences, both subdivisions are assumed to share the same cortical output system mediated by LV neurons. Recently, we and others have shown that LV in both MEC and LEC can

be genetically and connectionally divided into two sublayers: a deep layer Vb (LVb), which contains neurons receiving projections from the hippocampus, and a superficial layer Va (LVa), which originates the main projections out to forebrain cortical and subcortical structures (*Ohara et al., 2018*; *Ramsden et al., 2015*; *Sürmeli et al., 2015*; *Wozny et al., 2018*). These results indicate that for the hippocampal-cortical dialogue to function we need to postulate a projection from LVb to LVa neurons. Although the existence of such a LVb-LVa circuit is supported by our previous study using transsynaptic viral tracing in rats (*Ohara et al., 2018*), experimental evidence for functional connectivity from LVb-to-Va in LEC and MEC is still lacking.

In the present study, we examined the presence of this hypothetical intrinsic EC circuit by using a newly generated LVb-specific transgenic (TG) mouse line obtained with an enhancer-driven gene expression (EDGE) approach (*Blankvoort et al., 2018*). To compare the LVb intrinsic circuit between LEC and MEC, we ran identical in vitro electrophysiological and optogenetical experiments in comparable dorsal portions of LEC and MEC. To our surprise, we found differences in the postulated intrinsic LVb-LVa pathway between the two entorhinal subdivisions: the connectivity is prominent in dorsal LEC but is apparently sparse in dorsal parts of MEC. In contrast, other intrinsic circuits from LVb to layers II and III (LII and LIII), which constitute hippocampal-entorhinal re-entry circuits, are very similar in both entorhinal subdivisions. Our data seem to suggest that the current view of the canonical hippocampal-cortical output circuit that allegedly is crucial for systems consolidation might need revision, though the functional impact of our findings awaits further in vivo studies.

## Results

### Characterization of LVb TG mouse line

Entorhinal LV can be divided into superficial LVa and deep LVb based on differences in cytoarchitectonics, connectivity and genetic markers such as Purkinje cell protein 4 (PCP4) and chicken ovalbumin upstream promoter transcription factor interacting protein 2 (Ctip2) (*Ohara et al., 2018*; *Sürmeli et al., 2015*; *Figure 1—figure supplement 1*, see Materials and methods for details). To target the entorhinal LVb neurons, we used a TG mouse line (MEC-13-53D) that was obtained with the EDGE approach (*Blankvoort et al., 2018*). In this TG line, the tetracycline-controlled transactivator (tTA, Tet-Off) is expressed under the control of a specific enhancer and a downstream minimal promoter. To visualize the expression patterns of tTA, this line was crossed to a reporter mouse line, which expresses mCherry together with GCaMP6 in a tTA-dependent manner.

In both LEC and MEC, mCherry-positive neurons were observed mainly in LVb (93.2% in LEC and 82.9% in MEC) and some in layer VI (LVI; 5.1% in LEC and 16.8% in MEC) but hardly in LVa (1.7% in LEC and 0.3% in MEC), and none in superficial layers (*Figure 1A–D*). The proportion of PCP4-positive LVb neurons that show tTA-driven labeling was 45.9% in LEC and 30.9% in MEC (*Figure 1E*). The tTA-driven labeling colocalized well with the PCP4 labeling (percentage of tTA-expressing neurons that were PCP4-positive was 91.7% in LEC and 99.3% in MEC; *Figure 1F*), highlighting the specificity of the line. In another experiment using a GAD67 TG line expressing green fluorescent protein (GFP), we showed that the percentage of double-labeled (PCP4+, GAD67+) neurons among total GAD67-positive neurons is very low in both LEC and MEC (4.3% and 2.3%, respectively, *Figure 1—figure supplement 2*). This percentage of double-labeled neurons was significantly lower than in the Ctip2-stained sample in both regions (18.1% for LEC and 7.2% for MEC). This result shows that PCP4 can be used as a marker for excitatory entorhinal LVb neurons. Occasionally PCP4-positive neurons were observed in what seems to be layer Va, where there is a lack of continuity in the cell layer as indicated by retrograde tracing (*Figure 1—figure supplement 1*). Although sparse, these 'misplaced' LVb neurons were also targeted in our TG mouse line. The MEC-13-53D is thus an attractive TG mouse line to target excitatory LVb neurons in both LEC and MEC.

### Morphological properties of LVa/LVb neurons in LEC and MEC

We next examined the morphological and electrophysiological properties of the LVb neurons in LEC and MEC in this TG mouse line. Targeted LVb neurons were labeled by injecting tTA-dependent adeno-associated virus (AAV) encoding GFP (AAV2/1-TRE-Tight-GFP) into either LEC or MEC and filled with biocytin during whole-cell patch-clamp recordings in acute slices (*Figure 1G*). Consistent with our histological result showing that this line targets excitatory cells, all recorded cells showed

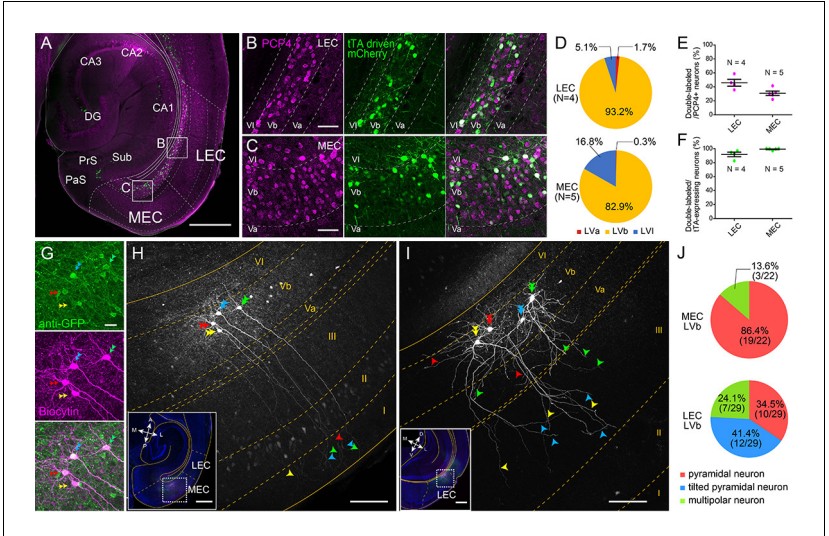

**Figure 1.** Lateral entorhinal cortex (LEC) and medial entorhinal cortex (MEC) layer Vb (LVb) neurons show distinct morphological features. (**A–C**) Expression of tetracycline-controlled transactivator (tTA) in the enhancer-driven gene expression (EDGE) mouse line (MEC-13-53D), which is visualized with mCherry (green) by crossing to a tTA-dependent mCherry line. A horizontal section was immunostained with an anti-Purkinje cell protein 4 (PCP4) antibody (magenta) to label entorhinal LVb neurons. Images of LEC (**B**) and MEC (**C**) correspond with the boxed areas in (**A**) and show from left to right PCP4 expression, mCherry expression, and a merged image. (**D**) Percentage of tTA-expressing neurons among layers in LEC and MEC. (**E**) Percentage of tTA-expressing neurons among the total PCP4-positive neurons in LEC and MEC. (**F**) Percentage of PCP4-positive neurons among the total tTA-expressing neurons in LEC and MEC. Error bars: mean ± standard errors. The tTA-expressing neurons mainly distributed in LVb of EC and colocalized with PCP4. (**G–I**) Morphology of LVb neurons targeted in MEC-13-53D in MEC (**G, H**) and LEC (**I**). tTA-expressing LVb neurons were first labeled with green fluorescent protein (GFP) (green) by injecting AAV2/1-TRE-Tight-EGFP in MEC-13-53D, and then intracellularly filled with biocytin (magenta, **G**) Images of MEC (**H**) and LEC (**I**) show biocytin labeling, which correspond with the boxed area in each inset. The four neurons shown in (**G**) correspond to the neurons in (**H**). Double arrowheads show the cell bodies, the single arrowheads show their dendrites, and different neurons are marked in different colors (green, blue, red, and yellow). The distribution of apical dendrites largely differs between MEC-LVb and LEC-LVb neurons. (**J**) Proportion of morphologically identified cell types of LVb neurons in LEC and MEC. These data were obtained in 10 animals and 22 slices. Scale bars represent 500 µm for (**A**) and inset of (**H**) and (**I**), 100 µm for (**H**) and (**I**), 50 µm for (**B**) and (**C**), and 20 µm for (**G**). *Figure 1—source data 1*. See also *Figure 1—figure supplement 1*, *Figure 1—figure supplement 2*, *Figure 1—figure supplement 3*, and *Figure 1—figure supplement 4*.

The online version of this article includes the following source data and figure supplement(s) for figure 1:

**Source data 1.** Specificity of tetracycline-controlled transactivator expression in MEC-13-53D.

**Figure supplement 1.** Laminar organization of lateral entorhinal cortex (LEC) and medial entorhinal cortex (MEC).

**Figure supplement 2.** Purkinje cell protein 4 (PCP4) but not chicken ovalbumin upstream promoter transcription factor interacting protein 2 (Ctip2) is expressed mainly in excitatory layer Vb (LVb) neurons in both lateral entorhinal cortex (LEC) and medial entorhinal cortex (MEC).

**Figure supplement 2—source data 1.** Specificity of Purkinje cell protein 4 and chicken ovalbumin upstream promoter transcription factor interacting protein 2 expression in entorhinal layer Vb neurons.

**Figure supplement 3.** Dendrites of lateral entorhinal cortex-layer Vb (LEC-LVb) neurons do not reach layer IIa and I.

**Figure supplement 4.** Medial entorhinal cortex (MEC) and lateral entorhinal cortex-layer Va (LEC-LVa) neurons share similar morphological features.

morphological and electrophysiological properties of excitatory neurons (*Figure 1*, *Figure 2*). In line with previous studies, many MEC-LVb neurons were pyramidal cells with apical dendrites that ascended straight toward layer I (LI; *Figure 1H, J*; *Canto and Witter, 2012a*; *Hamam et al., 2000*). In contrast, more than 40% of the targeted LEC-LVb neurons were tilted pyramidal neurons (*Canto and Witter, 2012b*; *Hamam et al., 2002*) with apical dendrites not extending superficially beyond LIII (*Figure 1I, J*). Since this latter result may result from severing of dendrites by the slicing procedure, we also examined the distribution of LVb apical dendrites in vivo. After injecting AAV2/

1-TRE-Tight-GFP in the deep layer of LEC in the TG line, the distribution of labeled dendrites of LEC-LVb neurons was examined throughout all sections (*Figure 1—figure supplement 3*). Even with this approach, the labeled dendrites mainly terminated in LIII and only sparsely reached layer IIb. These morphological differences indicate that MEC-LVb neurons sample inputs from different layers than LEC-LVb neurons: MEC-LVb neurons receive inputs throughout all layers, whereas LEC-LVb neurons only receive inputs innervating layer IIb–VI. In contrast to LVb neurons, the morphology of LVa neurons was relatively similar in both regions: the basal dendrites extended horizontally mostly within LVa, whereas the apical dendrites reached LI (*Figure 1—figure supplement 4*). These morphological features of LVa neurons are in line with previous studies (*Canto and Witter, 2012a*; *Canto and Witter, 2012b*; *Hamam et al., 2000*; *Hamam et al., 2002*; *Sürmeli et al., 2015*).

## Electrophysiological properties of LVa/LVb neurons in LEC and MEC

Previous studies have reported that the electrophysiological profiles of LV neurons are diverse both in LEC and MEC (*Canto and Witter, 2012a*; *Canto and Witter, 2012b*; *Hamam et al., 2000*; *Hamam et al., 2002*), but whether these different electrophysiological properties of entorhinal LV neurons relate to the two sublayers, LVa and LVb, was unclear. Here, we examined this by analyzing a total of 121 neurons recorded from the TG mouse line (MEC-13-53D): 31 LEC-LVa, 45 LEC-LVb, 20 MEC-LVa, and 25 MEC-LVb neurons (*Figure 2A*). As previously reported (*Canto and Witter, 2012a*; *Canto and Witter, 2012b*; *Hamam et al., 2000*; *Hamam et al., 2002*), only a few LV neurons showed weak depolarizing afterpotentials (DAP; *Figure 2B*), with a higher incidence in MEC than in LEC. Among the 12 examined electrophysiological properties (*Figure 2—source data 1*), differences were observed between the LVa and LVb neurons in most parameters except for resting potential, input resistance, and action potential (AP) threshold (*Figure 2F–K*, *Figure 2—figure supplement 1*). Principal component analysis based on the 12 parameters resulted in a clear separation between LVa and LVb neurons, and also in a moderate separation between LEC-LVb and MEC-LVb (*Figure 2L*). Sag ratio (*Figure 2C, F*), time constant (*Figure 2G*), and AP frequency after 200 pA injection (*Figure 2E, H*) were the three prominent parameters that separated LVa and LVb neurons (*Figure 2M*) with the sag ratio and AP frequency after 200 pA injection being smaller in LVa than LVb, and the opposite was true for the time constant. The difference in sag ratio may indicate that LVb neurons show more prominent subthreshold oscillations, which have been reported to occur in LV, although details on differences between the two sublayers have not been studied (*Egorov et al., 2002b*; *Schmitz et al., 1998*). The clearest features aiding in separating LEC-LVb and MEC-LVb were time constant (*Figure 2G*), AP frequency after 200 pA injection (*Figure 2E, H*), and fast afterhyperpolarization (AHP; *Figure 2I, N*). Neurons in MEC-LVb showed a smaller time constant, higher AP frequency, and smaller fast AHP than LEC-LVb neurons. Although LVb neurons in LEC and MEC thus differed in some of their electrophysiological characteristics, as well as morphologically (described above; *Figure 1G–J*), it remains to be determined how these two features influence neuronal and network activity.

## Local projections of LVb neurons in LEC and MEC are different

Subsequently, we examined the local entorhinal LVb circuits by injecting a tTA-dependent AAV carrying both the channelrhodopsin variant oChIEF and the yellow fluorescent protein (citrine, AAV2/1-TRE-Tight-oChIEF-citrine) into the deep layers of either LEC or MEC in mouse line MEC-13-53D (*Figure 3A*). This enabled specific expression of the fused oChIEF-citrine protein in either LEC-LVb (*Figure 3B–F*) or MEC-LVb (*Figure 3G–K*). Not only the dendrites and the soma but also the axons of these LVb neurons were clearly labeled. As shown in the horizontal sections taken at different dorsoventral level (*Figure 3B–D, G–I*), citrine-labeled axons were observed mainly within the EC, and only very sparse labeling was observed in other regions, including the angular bundle, a major efferent pathway of EC. This result supports our previous study (*Ohara et al., 2018*) showing that the main targets of the entorhinal LVb neurons are neurons in superficially positioned layers. Within EC, the distribution of labeled axons differed between LEC and MEC (*Figure 3L*). Although in both LEC and MEC, labeled axons were densely present in LIII rather than in layers II and I, we report a striking difference between LEC and MEC in LVa, as is easily appreciated from *Figure 3L, M*: many labeled axons of LEC-LVb neurons were present in LVa, whereas in case of MEC-LVb, the number of labeled axons was very low in LVa. Such entorhinal labeling patterns were not affected by the unintended

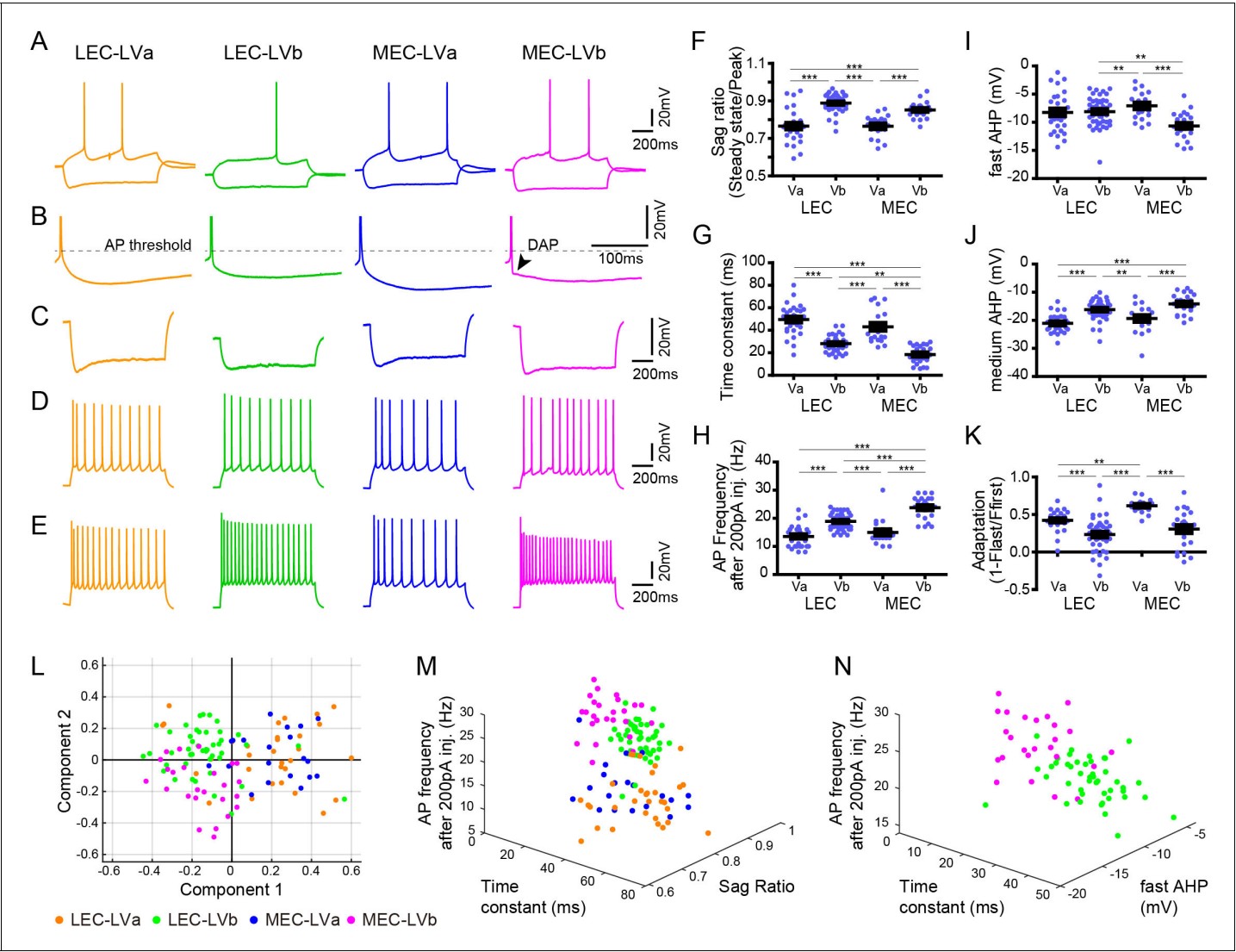

**Figure 2.** Electrophysiological properties distinguish layer Va/layer Vb (LVa/LVb) neurons in both lateral entorhinal cortex (LEC) and medial entorhinal cortex (MEC). (A) Representative voltage responses to hyperpolarizing and depolarizing current injection of LEC-LVa (orange), LEC-LVb (green), MEC-LVa (blue), and MEC-LVb (magenta) neurons. (B) Voltage responses at rheobase current injections showing afterhyperpolarization (AHP) wave form and depolarizing afterpotentials (DAP). (C) Voltage responses to hyperpolarizing current injection with peaks at −90 ± 5 mV showing Sag. (D) Voltage responses to depolarizing current injection with 10 ± 1 action potentials (APs) showing adaptation. (E) Voltage responses to +200 pA of 1-s-long current injection showing maximal AP number. (F–K) Differences of sag ratio (F, one-way ANOVA, $F_{3,117}$ = 36.88, ***p<0.0001, Bonferroni's multiple comparison test, ***p<0.001), time constant (G, one-way ANOVA, $F_{3,117}$ = 53.39, ***p<0.0001, Bonferroni's multiple comparison test, **p<0.01, ***p<0.001), AP frequency after 200 pA injection (H, one-way ANOVA, $F_{3,117}$ = 44.37, ***p<0.0001, Bonferroni's multiple comparison test, ***p<0.001), fast AHP (I, one-way ANOVA, $F_{3,117}$ = 7.536, ***p=0.0001, Bonferroni's multiple comparison test, **p<0.01, ***p<0.001), medium AHP (J, one-way ANOVA, $F_{3,117}$ = 21.99, ***p<0.0001, Bonferroni's multiple comparison test, **p<0.01, ***p<0.001), and adaptation (K, one-way ANOVA, $F_{3,117}$ = 21.6, ***p<0.0001, Bonferroni's multiple comparison test, **p<0.01, ***p<0.001) between LEC-LVa (N = 31), LEC-LVb (N = 45), MEC-LVa (N = 20), and MEC-LVb (N = 25) neurons (error bars: mean ± standard errors). (L) Principal component analysis based on the 12 electrophysiological parameters shown in *Figure 2—source data 1* show a separation between LVa and LVb neurons as well as a moderate separation between LEC-LVb and MEC-LVb neurons. Data representing 121 neurons from 27 animals (also holds for M and N). (M) Separation of LEC-LVa (orange), LEC-LVb (green), MEC-LVa (blue), and MEC-LVb (magenta) neurons using sag ratio, AP frequency at 200 pA injection, and time constant as distinction criteria. (N) Separation of LEC-LVb (green) and MEC-LVb (magenta) neurons using fast AHP, AP frequency at 200 pA injection, and time constant as distinction criteria.

The online version of this article includes the following source data and figure supplement(s) for figure 2:

**Source data 1.** Electrophysiological properties of entorhinal layer V neurons.

**Figure supplement 1.** Electrophysiological features of layer Va/layer Vb (LVa/LVb) neurons in lateral entorhinal cortex/medial entorhinal cortex (LEC/MEC).

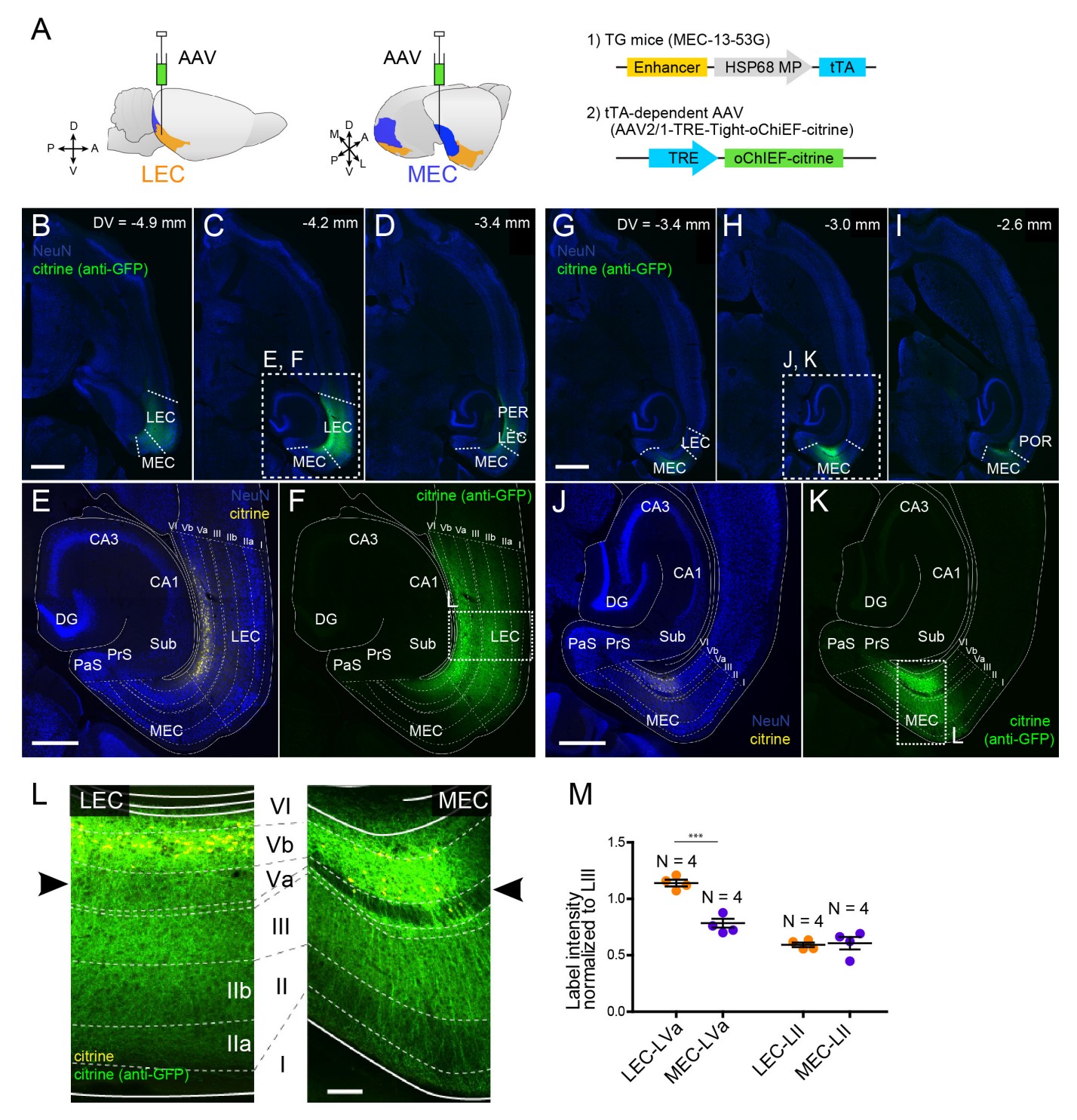

**Figure 3.** Layer Vb (LVb) neurons project locally, and their projections differ between lateral entorhinal cortex (LEC) and medial entorhinal cortex (MEC). (A) Tetracycline-controlled transactivator (tTA)-expressing entorhinal LVb neurons were visualized by injecting a tTA-dependent adeno-associated virus (AAV) expressing oChIEF-citrine into either LEC or MEC of MEC-13–53G. (B–F) Horizontal sections showing distribution of labeled neurites originating from LEC-LVb at different dorsoventral levels (B–D). Images of entorhinal cortex (EC) (E, F) correspond to the boxed area in (C). Note that the cell bodies of labeled neurons are located in LVb of LEC (citrine label in yellow, E), and that the labeled neurites mainly distribute within EC (citrine immunolabeling in green, F). The labeling observed in perirhinal cortex (PER; D) originates from the sparse infection of PER neurons due to the leakage of the virus along the injection tract. (G–K) Horizontal sections showing distribution of labeled fibers originating from MEC-LVb at different dorsoventral levels (G–I). Images of EC (J, K) correspond to the boxed area in (H). Note that the cell bodies of labeled neurons are located in LVb of MEC (J), and

*Figure 3 continued on next page*

Figure 3 continued

that the labeled neurites mainly distribute within EC (K). The labeling observed in postrhinal cortex (POR; I) originates from the sparse infection of POR neurons due to the leakage of the virus along the injection tract. (L) Comparison of labeled neurites originating from LEC-LVb and MEC-LVb neurons (green), of which the cell bodies are visualized with citrine (yellow). The left panel corresponds to the boxed area in (F) and is 90° rotated to match the orientation of the right panel, which represents the boxed area in (K). The distribution of the labeled fibers is strikingly different between LEC and MEC in LVa (black arrowhead) with a strong terminal projection in LEC and almost absent projections in LVa of MEC. (M) Intensity of citrine immunolabeling in LVa and LII of LEC and MEC, normalized against the LIII labeling (error bars: mean ± standard errors, N = 4). The normalized labeling was significantly higher in LEC-LVa than in MEC-LVa (two-tailed paired t-test for LEC-LVa vs. MEC-LVa: t6 = 7.68, \*\*\*p=0.0003, LEC-LII vs. MEC-LII: t6 = 0.24, p=0.82). Scale bars represent 1000 μm for (B) and (G) (also apply to C, D, H, I), 500 μm for (E) and (J) (also apply to F and K), and 100 μm for (L).

*Figure 3—source data 1*. See also *Figure 3—figure supplement 1*.

The online version of this article includes the following source data and figure supplement(s) for figure 3:

**Source data 1.** Distribution of labeled fibers of layer Vb neurons in entorhinal layer V.

**Figure supplement 1.** Axonal distribution of perirhinal cortex (PER), postrhinal cortex (POR), and entorhinal layer Vb (LVb) neurons.

labeled neurons in the deep perirhinal cortex (PER; *Figure 3D*) or postrhinal cortex (POR; *Figure 3I*) since these neurons hardly project to LEC and MEC (*Figure 3—figure supplement 1A–C*). It is also very unlikely that the sparse labeling patterns in MEC-LVa is a false negative result due to the selective targeting of a LVb subpopulation that modestly project to LVa for two reasons. First, the PCP4-labeling patterns referred to above also differ in LVa between MEC and LEC: PCP4-labeled fibers are hardly present in MEC LVa, whereas the axonal density is much higher in LEC LVa (*Figure 1—figure supplement 1*). Second, a strikingly similar labeling pattern was observed in LVa of MEC following an anterograde tracer injection into MEC-LVb in wild-type mice (*Figure 3—figure supplement 1D*). Note that we also confirmed this labeling pattern in rat MEC (*Figure 3—figure supplement 1E*), which is in line with a previous study (*Köhler, 1986*). Based on these anatomical observations, we predicted that LVb neurons in both LEC and MEC innervate LIII neurons rather than LII neurons. Importantly, our findings further indicate that LVb-to-LVa connections, which mediate the hippocampal-cortical output circuit, are much more prominent in LEC than in MEC. To test these predicted connectivity patterns, we used optogenetic stimulation of the oChIEF-labeled axons together with patch-clamp recordings of neurons in the different layers of EC.

## Translaminar local connections of MEC-LVb neurons

We first examined the LVb circuits in MEC by performing patch-clamp recording from principal neurons in layers II (n = 20 for stellate cells, n = 18 for pyramidal cells), III (n = 30), and Va (n = 18), while optically stimulating LVb fibers in acute horizontal entorhinal slices (*Figure 4*, *Figure 4—figure supplement 1*). Recorded neurons were labeled with biocytin, and the neurons were subsequently defined from the location of their cell bodies, morphological characteristics, and electrophysiological properties. In line with previous studies, LIII principal neurons were pyramidal cells, while neurons in LII were either stellate cells or pyramidal neurons (*Figure 4A*; *Canto and Witter, 2012a*; *Fuchs et al., 2016*; *Winterer et al., 2017*). LII stellate cells were not only identified by the morphological features but also from their unique physiological properties, characterized by the pronounced sag potential and DAP (*Figure 4B*; *Alonso and Klink, 1993*).

There was a densely labeled axonal plexus in LIII, which is the layer where LIII pyramidal neurons mainly distribute their basal dendrites. In line with this anatomical observation, all LIII neurons (30 out of 30 cells) responded to the optical stimulation (*Figure 4C, D*). In contrast, the axonal labeling was sparse in LII, and this distribution was reflected in the observed sparser connectivity. The percentage of pyramidal neurons in LII responding to optical stimulation was 61.1% (11 out of 18 cells), and this percentage was especially low in stellate cells (25.0%; 5 out of 20 cells). Even in the five stellate cells that responded to the light stimulation, evoked responses were relatively small as measured by the amplitude of the synaptic event (*Figure 4C, E*, *Figure 4—figure supplement 1C, E*). In order to compare the differences of excitatory postsynaptic potential (EPSP) amplitudes across different layers/cell types, the voltage responses of each neuron were normalized to the response of LIII cells recorded in the same slice (*Figure 4F*). The normalized EPSP amplitude of LII cells was significantly smaller than those of LIII pyramidal cells, and within LII cells, the normalized responses of stellate cells were significantly smaller than those of pyramidal cells (LII stellate cells, 0.24 ± 0.06; LII pyramidal cells, 0.63 ± 0.08; LIII pyramidal cells, 1.0 ± 0.03; p<0.001 for LIIs vs. LIII and LIIp vs. LIII,

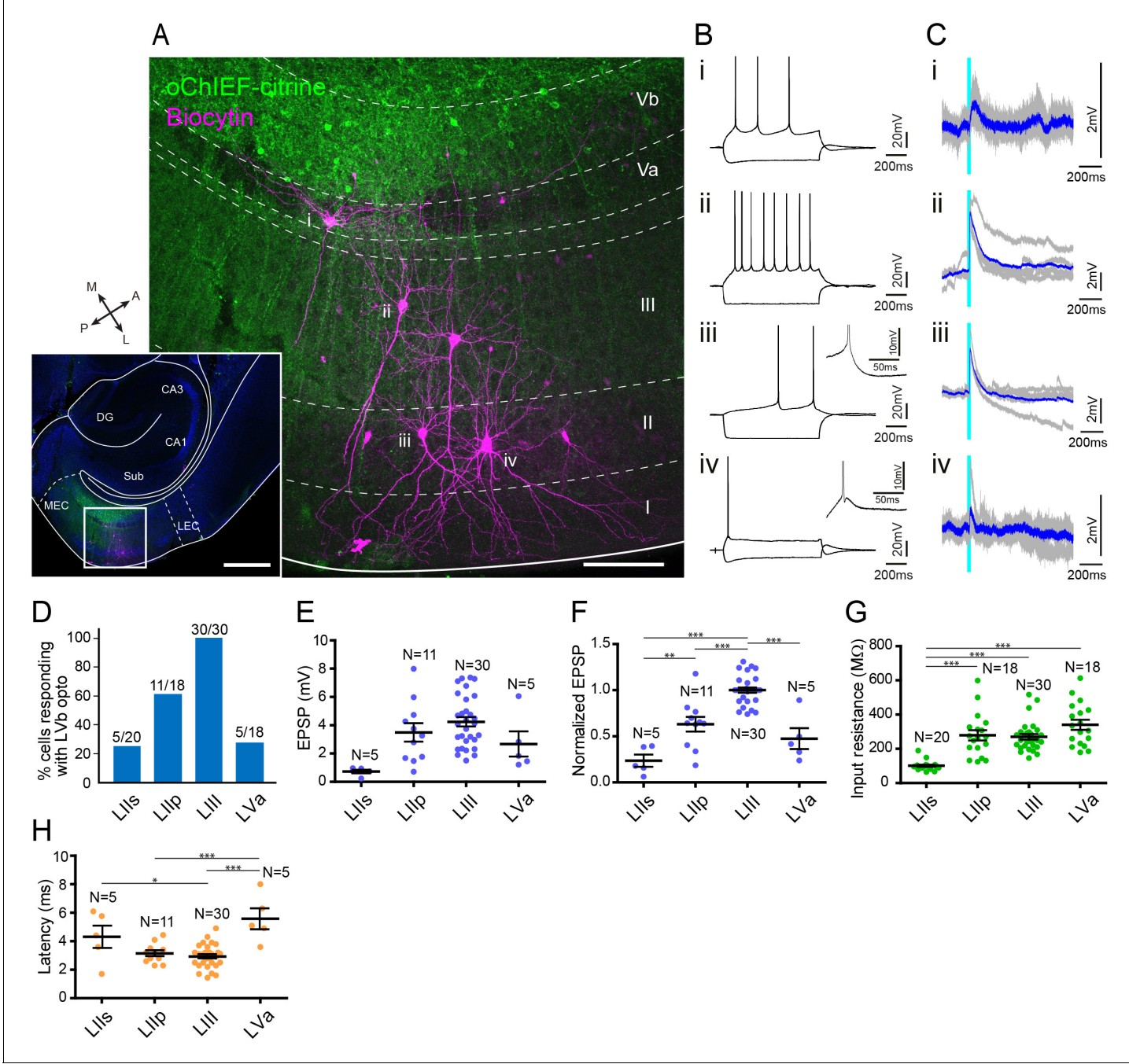

**Figure 4.** Medial entorhinal cortex-layer Vb (MEC-LVb) neurons preferentially target LII/III pyramidal neurons. (A) Image of a representative horizontal slice showing expression of oChIEF-citrine in LVb neurons (green) and recorded neurons labeled with biocytin (magenta) in MEC. Inset shows a low-power image of the section indicating the position of the higher power image. Scale bars represent 500 μm (inset) and 100 μm. (B) Voltage responses to injected current steps recorded from neurons shown in (A): i, pyramidal cell in layer Va (LVa); ii, pyramidal cell in LIII; iii, pyramidal cell in LII; iv, stellate cell in LII. Inset in (iii) and (iv) shows the depolarizing afterpotential (DAP) in expanded voltage- and time scale. Note that LII stellate cells (iv) show a clear sag potential and DAP compared to LII pyramidal cells (iii). (C) Voltage responses to light stimulation (light blue line) recorded from neurons shown in (A). Average traces (blue) are superimposed on the individual traces (gray). (D–G) The proportion of responding cells (D), excitatory postsynaptic potential (EPSP) amplitude (E), the normalized EPSP based on LIII response (F, one-way ANOVA, $F_{3,47}$ = 33.29, ***p<0.0001, Bonferroni's multiple comparison test, **p<0.01, ***p<0.001), and the input resistance (G, one-way ANOVA, $F_{3,82}$ = 21.99, ***p<0.0001, Bonferroni's multiple comparison test, ***p<0.001) was examined for each cell type (error bars: mean ± standard errors). (H) Latency of EPSP onset for MEC neurons to optical activation (F, one-way ANOVA, $F_{3,47}$ = 11.65, ***p<0.0001, Bonferroni's multiple comparison test, *p<0.05, ***p<0.001). LIIs: LII stellate cell; LIIp: LII pyramidal cell. *Figure 4—source data 1*. See also *Figure 4—figure supplement 1*, *Figure 4—figure supplement 2*.

*Figure 4 continued on next page*

*Figure 4 continued*

The online version of this article includes the following source data and figure supplement(s) for figure 4:

**Source data 1.** Patch-clamp recording data in medial entorhinal cortex.
**Figure supplement 1.** Representative patch-clamp recordings after optical stimulation of layer Vb (LVb) fibers in medial entorhinal cortex (MEC).
**Figure supplement 2.** Responses of medial entorhinal cortex-layer Va (MEC-LVa) neurons at different dorsoventral levels.
**Figure supplement 2—source data 1.** Responses of medial entorhinal cortex-layer Va neurons at different dorsoventral levels.

p<0.01 for LIIs vs. LIIp, one-way ANOVA followed by Bonferroni's multiple comparison test). This difference in responses between LII and III neurons is likely due to the difference of the LVb fiber distribution within these layers. In contrast, the difference of responses between the two cell types in LII might be explained by the difference of the input resistance between these neurons, the input resistances of stellate cells being significantly lower than that of pyramidal cells (101.2 ± 5.9 vs. 278.9 ± 29.1 MΩ; p<0.001, one-way ANOVA followed by Bonferroni's multiple comparison test; *Figure 4G*), although we cannot exclude a possible difference in total synaptic input between the stellate and pyramidal neurons in LII. Note that the latency of the EPSP onset for LII stellate cells was significantly longer than that for LIII cells (4.3 ± 0.7 vs. 2.9 ± 0.1 ms; p<0.5, one-way ANOVA followed by Bonferroni's multiple comparison test; *Figure 4H*). In contrast, the inputs to LII pyramidal cells showed latencies similar to those of LIII cells (3.2 ± 0.2 vs. 2.9 ± 0.1 ms). These differences in latencies indicate that projections from LVb to LIII and LII pyramidals may be monosynaptic and those to LII neurons may be disynaptic. Since we did not assess this pharmacologically, it is hard to conclude, particularly since in a parallel study in mouse LEC LII, using similar viral and laser stimulation protocols, we measured latencies ranging from 4.5 to 6.8 ms, which consistently were shown to be monosynaptic when tested using pharmacological measures (*Nilssen, 2019*).

As shown in *Figure 1—figure supplement 4*, and also in line with previous studies (*Canto and Witter, 2012a*; *Hamam et al., 2000*; *Sürmeli et al., 2015*), LVa pyramidal cells have their basal dendrites mainly confined to LVa, which is the layer that MEC-LVb neurons avoid to project to (*Figure 3L*, *Figure 4A*, *Figure 4—figure supplement 1D, G*). In line with this anatomical observation, only 27.8% (5 out of 18 cells) responded to the light stimulation (*Figure 4D*). The EPSP amplitudes of the responding LVa neurons were relatively small (*Figure 4C, E*, *Figure 4—figure supplement 1F, H*), and the normalized EPSPs were significantly smaller than those of LIII neurons (0.47 ± 0.10 vs. 1.0 ± 0.03; p<0.001, one-way ANOVA followed by Bonferroni's multiple comparison test; *Figure 4F*). In addition, the latency of the EPSP onset for LVa cells was significantly longer than that for LIII cells (5.6 ± 1.7 vs. 2.9 ± 0.1 ms; p<0.001, one-way ANOVA followed by Bonferroni's multiple comparison test; *Figure 4H*), indicating that these responses are either the result of monosynaptic inputs onto the apical dendrite in LIII or that they represent disynaptic responses.

We recorded in slices taken at different dorsoventral levels. Since functional differences along this axis have been reported (*Steffenach et al., 2005*; *Stensola et al., 2012*), we examined whether the LVb-LVa connectivity differs along the dorsoventral axis by grouping the recorded LVa responses in three distinct dorsoventral levels (*Figure 4—figure supplement 2*). The voltage responses of the more ventrally positioned LVa neurons were significantly higher than those measured more dorsally in MEC (p<0.01, one-way ANOVA followed by Bonferroni's multiple comparison test). Since the EPSP amplitudes of LIII neurons did not differ at different dorsoventral levels, it is unlikely that the observed response differences are caused by different levels of oChIEF expression in LVb fibers along the dorsoventral axis. The observed difference may be caused by the difference in severing of apical dendrites at different dorsoventral levels since the axons of MEC-LVb neurons massively distribute in LIII and frequently reach into LI. This, however, does not seem to be the case since LVa cells in the dorsal MEC, irrespective of whether they have full (n = 3) or severed dendrites (n = 2), did not respond to light stimulation. Similarly, for the central dorsoventral level, four out of five neurons with intact apical dendrites in LIII did not respond. In the ventral MEC recordings, four out of five neurons with seemingly intact dendrites responded to the optical stimulation (*Figure 4—figure supplement 2D*).

## Translaminar local connections of LEC-LVb neurons

We next examined the LVb local circuits in LEC with the similar method as applied in MEC (above). In LEC, LII can further be divided into two sublayers: a superficial layer IIa (LIIa) composed of fan cells and a deep layer IIb (LIIb) mainly composed of pyramidal neurons (*Leitner et al., 2016*). Fan cells mainly extend their apical dendrites in LI, where the density of LVb labeled fibers is extremely low (*Figure 5A*, *Figure 5—figure supplement 1*). This contrasts with LIIb, LIII, and LVa neurons, which distribute at least part of their dendrites in layers with a relatively high density of LVb axons. In line with these anatomical observations, only 26.9% of the fan cells (7 out of 26 neurons) responded to the light stimulation (*Figure 5C, D*). On the other hand, the response probabilities of LIIb, III, and Va were high, 76.9% (20 out of 26 cells), 100% (34 out of 34 cells), and 94.7% (18 out of 19 cells), respectively. The voltage responses of these neurons were also significantly larger than those of the LIIa neurons (LIIa neurons, $0.15 \pm 0.03$; LIIb neurons, $0.79 \pm 0.15$; LIII neurons, $1.0 \pm 0.04$; LVa neurons, $0.93 \pm 0.10$; $p<0.01$ for LIIa vs. LIIb, $p<0.001$ for LIIa vs. LIII and LIIa vs. LVa, one-way ANOVA followed by Bonferroni's multiple comparison test; *Figure 5E, F*). In contrast to MEC-LII stellate cells, the input resistance of LIIa fan cells was significantly higher than that of LIIb and LIII neurons (LIIa neurons, $323.6 \pm 16.1$; LIIb neurons, $226.5 \pm 17.2$; LIII neurons, $194.3 \pm 13.3$ MΩ; $p<0.001$, one-way ANOVA followed by Bonferroni's multiple comparison test; *Figure 5G*). This indicates that the small responses of LIIa fan cells cannot be explained by the differences in input resistance among the superficial neurons and may simply be due to the small number of synaptic inputs to LIIa fan cells from LVb neurons. In contrast to MEC, LEC-LVa neurons showed large responses to light stimulation, which matched with the anatomically dense LVb fiber distribution in LEC-LVa (*Figure 3L*). In line with our previous monosynaptic input tracing study using rabies virus (*Ohara et al., 2018*), the latency of the EPSP onset, which was similar in all cell types (LIIa neurons, $4.8 \pm 1.1$; LIIb neurons, $5.0 \pm 1.4$; LIII neurons, $4.2 \pm 0.9$; LVa neurons, $4.2 \pm 0.6$ ms; *Figure 5H*), points to the LVb-to-LVa connectivity, as well as that to LIII and LII, being largely monosynaptic.

The striking difference between MEC and LEC regarding LVb to LVa projections is clear from comparing the proportion of responding neurons (*Figure 5I*), and the normalized EPSP based on LIII response (*Figure 5J*) between MEC and LEC. In contrast to the similar responses of LII neurons between the two subregions, the normalized voltage responses of LEC-LVa neurons were significantly larger than those of MEC-LVa neurons ($0.93 \pm 0.10$ vs. $0.47 \pm 0.10$, $p<0.05$, two-tailed unpaired t-test; *Figure 5J*). On a final note, it is apparent that postsynaptic responses in LEC are larger than those in MEC. This difference in response amplitudes is most likely due to the differences in the number of oChIEF-expressing neurons since, as shown in *Figure 1E*, the proportion of tTA-expressing neurons is higher in LEC than in MEC. We deem it unlikely that these amplitude differences are caused by differences in biophysical properties since no such differences have been reported between matching cell types in LEC and MEC (*Canto and Witter, 2012a*). We also did not observe striking effects of the maintenance of the apical dendritic tree of LEC-LVa; neurons showed responses irrespective of whether they had dendrites restricted to layer V (n = 14) or additionally showed dendrites extending into layer III (n = 5; *Figure 4—figure supplement 2D*).

The present data clearly show that neurons in LVb of both LEC and MEC give rise to dense intrinsic projections to more superficial layers and show laminar preferences (*Figure 6*). We noticed a striking difference between the two entorhinal regions, in that neurons in LEC-LVb seemed to innervate LVa neurons, whereas in dorsal MEC this was rarely the case. In contrast, other intrinsic circuits from LVb to LII/III were very similar in both entorhinal subdivisions, which preferentially targeted pyramidal cells rather than the stellate or fan cells in LII. These data indicate that the weak experimental support for a strong LVb-to-LVa projection in dorsal MEC is not due to technical issues. Rather, the data provide evidence that the intrinsic LV circuitry that supposedly underlies the canonical hippocampal-cortical output circuit in dorsal MEC is differently organized than in other parts of EC.

## Discussion

In this study, we experimentally tested the major assumption about the organization of hippocampal-cortical output circuits via entorhinal LVb neurons, considered to be crucial for the normal functioning of the medial temporal lobe memory system, more in particular systems memory consolidation. Our key finding is that LEC and MEC are strikingly different in that excitatory

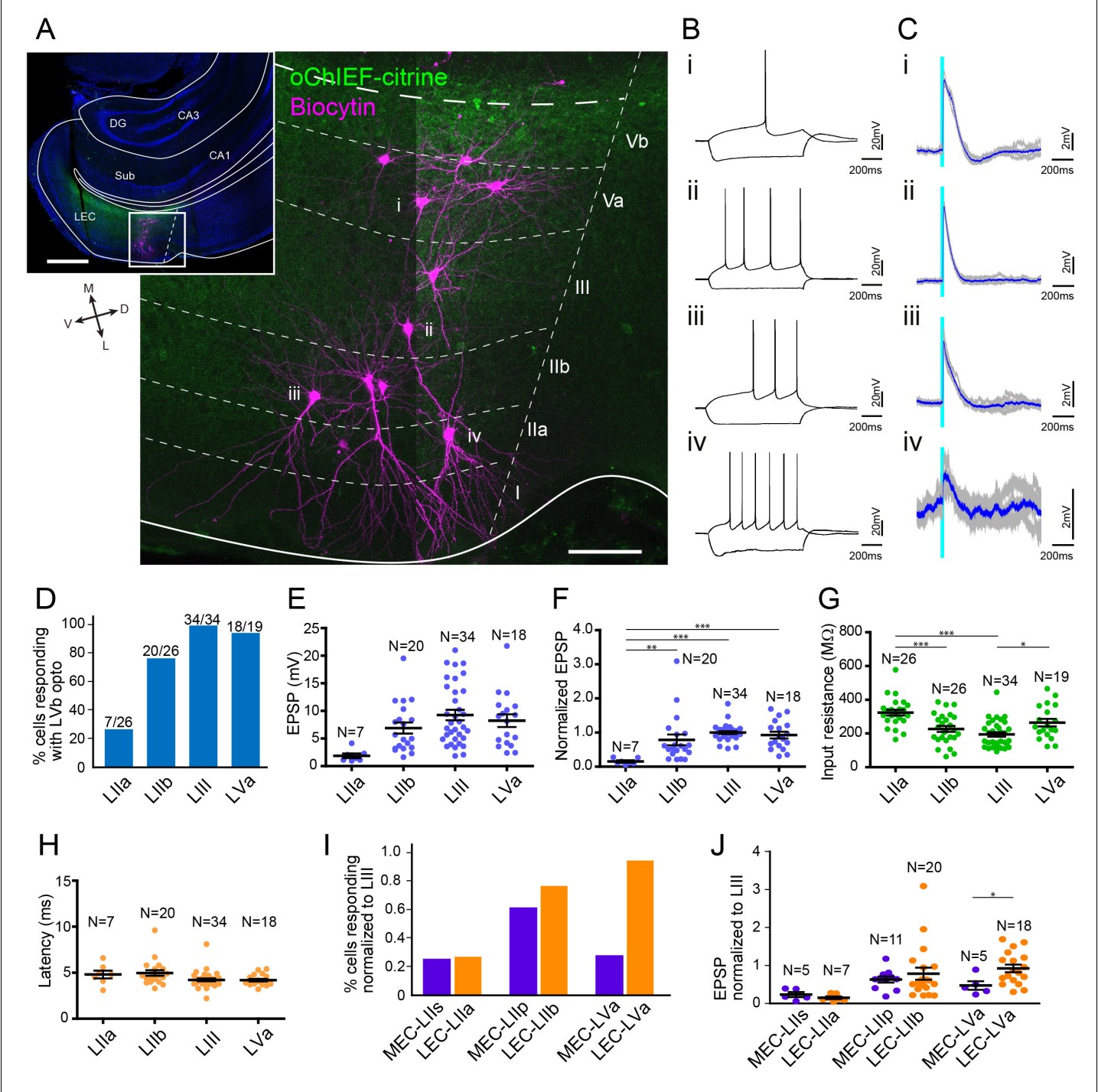

**Figure 5.** Lateral entorhinal cortex-layer Vb (LEC-LVb) neurons target layer Va (LVa) pyramidal neurons as well as LII/III pyramidal neurons. (**A**) Representative image of semicoronal slice showing expression of oChIEF-citrine in LVb neurons (green) and recorded neurons labeled with biocytin (magenta) in LEC. Inset shows a low-power image of the section indicating the position of the higher-power image. Scale bars represent 500 μm (inset) and 100 μm. (**B**) Voltage responses to injected current steps recorded from neurons shown in (**A**): i, pyramidal cell in LVa; ii, pyramidal cell in LIII; iii, pyramidal cell in LII; iv, fan cell in LII. (**C**) Voltage responses to light stimulation (light blue line) recorded from neurons shown in (**A**). Average traces (blue) are superimposed on the individual traces (gray). (**D–G**) The proportion of responding cells (**D**), excitatory postsynaptic potential (EPSP) amplitude (**E**), the normalized EPSP based on LIII response (**F**, one-way ANOVA, $F_{3,75}$ = 7.675, ***p=0.0002, Bonferroni's multiple comparison test, **p<0.01, ***p<0.001), and the input resistance (**G**, one-way ANOVA, $F_{3,101}$ = 11.75, ***p<0.0001, Bonferroni's multiple comparison test, *p<0.05, ***p<0.001) was examined for each cell type (error bars: mean ± standard errors). (**H**) Latency of EPSP onset for LEC neurons to optical activation (one-way ANOVA, $F_{3,47}$ = 11.65). (**I, J**) Comparison of the proportion of responding cells (**I**) and the normalized EPSP based on LIII response (**J**) between medial entorhinal

*Figure 5 continued on next page*

*Figure 5 continued*

cortex (MEC) and LEC (error bars: mean ± standard errors; two-tailed unpaired t-test, $t_{21}$ = 2.239, *p=0.0361). *Figure 5—source data 1*. See also *Figure 5—figure supplement 1*.

The online version of this article includes the following source data and figure supplement(s) for figure 5:

**Source data 1.** Patch-clamp recording data in lateral entorhinal cortex.

**Figure supplement 1.** Representative patch-clamp recording after optical stimulation of layer Vb (LVb) fibers in lateral entorhinal cortex (LEC).

connectivity from LVb to LVa is anatomically denser and electrophysiologically stronger in dorsal LEC than in dorsal MEC. In addition, we present new data that point to three major functionally relevant insights in the organization of the intrinsic translaminar entorhinal network originating from LVb neurons.

First, the present data indicate that LVb pyramidal neurons in LEC and MEC differ with respect to main morphological and electrophysiological characteristics. In contrast, LVa neurons in MEC and LEC are rather similar in these two aspects. Second, we show that projections from principal neurons in LVb in both entorhinal subdivisions preferentially contact pyramidal neurons in LIII and LII. LVb neurons have a sparser connectional relationship with principal neurons in LII that project to the dentate gyrus (DG) and the CA3/CA2 region, i.e., stellate and fan cells. Last, and most important, our data point to a new and challenging circuit difference between the two entorhinal subdivisions with respect to the inputs to LVa neurons, i.e., the output neurons of EC. Whereas in LEC, LVa neurons receive substantial input from LVb neurons, this projection is relatively weak in dorsal MEC. This difference in dorsal MEC, though unexpected in view of previous data including our own rabies tracing data (*Ohara et al., 2018*), has been recently corroborated in an in vitro study using paired-patch recording (*Rozov et al., 2020*), and our present corroborating tracing data are in line with previous tracing data in the rat (*Köhler, 1986*) and monkey (*Chrobak and Amaral, 2007*).

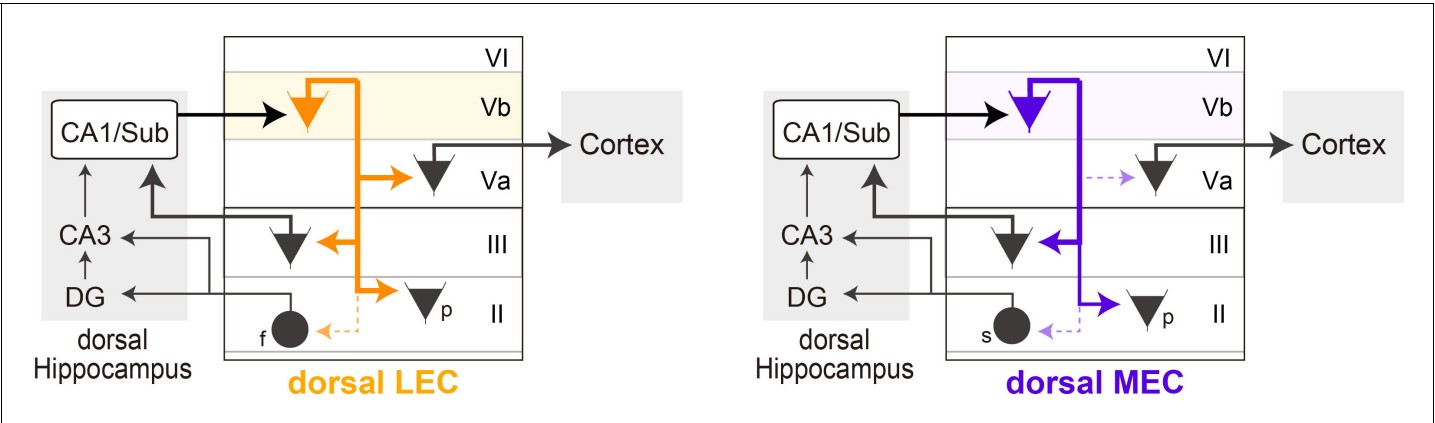

**Figure 6.** Schematic diagram of the different local circuits in lateral entorhinal cortex (LEC) and medial entorhinal cortex (MEC) used by layer Vb (LVb) neurons to transfer dorsal hippocampal output. Local connectivity of LVb neurons in LEC (left, orange) and MEC (right, purple). In both LEC and MEC, LVb neurons are the primary recipients of dorsal hippocampal output, but the transfer to LVa neurons through direct LVb-to-LVa projections is only prominent in LEC. Such projections are sparse and weak in MEC. Neurons in LVa are the output neurons of EC, projecting to the neocortex and other telencephalic subcortical structures. In contrast, in both LEC and MEC we find projections from LVb that target pyramidal cells in LIII, including neurons projecting to CA1 and subiculum, and pyramidal cells in LII. Projections to stellate (MEC) and fan (LEC) cells, which project to the dentate gyrus and CA3, are sparse and weak. The output projections of LII pyramidal neurons are not indicated in the figure, they project to ipsilateral-EC, contralateral-EC, CA1, or other telencephalic structures (*Ohara et al., 2019*). For clarity reasons, all these projections are indicated schematically as originating from a single LVb neuron, but this is not yet known. f: fan cell; s: stellate cell; p: pyramidal cell.

## Layer Vb neurons in LEC and MEC are morphologically and electrophysiologically different

The use of layer-specific TG mice allowed us to differentiate neurons in LVa from those in LVb and to differentiate these layer-specific neuron types between LEC and MEC. This contrasts with previous studies in rats, showing that LV neurons in both LEC and MEC share electrophysiological properties (*Canto and Witter, 2012a*; *Canto and Witter, 2012b*; *Hamam et al., 2000*; *Hamam et al., 2002*), although these authors did differentiate between LVa and LVb neurons based on morphological criteria and laminar distribution. We corroborate the reported morphological differences and add that the neurons also differ with respect to their electrophysiological properties. The most striking difference between MEC- and LEC-LVb neurons, however, is in the morphology of the apical dendrite. Neurons in MEC-LVb have an apical dendrite that heads straight to the pia, such that distal branches reach all the way up into LI, which is in line with previous studies (*Canto and Witter, 2012a*; *Hamam et al., 2000*; *Sürmeli et al., 2015*). In contrast, the apical dendrites of LEC-LVb neurons have a more complex branching pattern and they do not extend beyond LIII. This indicates that LEC-LVb neurons are unlikely to be targeted by inputs to LEC that selectively distribute to layers I and II, such as those carrying olfactory information from the olfactory bulb and the piriform cortex (*Luskin and Price, 1983*) as well as commissural projections (*Leitner et al., 2016*). The LVb neurons in LEC are thus dissimilar to their counterparts in MEC, which are morphologically suited to receive such superficially terminating inputs, as has been shown for inputs from the parasubiculum (*Canto et al., 2012*) and contralateral MEC (*Fuchs et al., 2016*). The here reported differences between LVb neurons, with MEC-LVb neurons showing a shorter time constant than LEC-LVb neurons, further indicate that MEC-LVb neurons have a shorter time window to integrate inputs compared to LEC-LVb neurons (*Canto and Witter, 2012a*; *Canto and Witter, 2012b*). The differences in AP frequency and half duration of AP may result in differences in the propensity of neurons to show graded persistent firing, which is prominent in MEC LV. Unfortunately, reports of persistent activity in MEC do not differentiate between neurons in LVa and LVb (*Egorov et al., 2002a*; *Fransén et al., 2006*). However, up-down state activity originating in LIII particularly entrains neurons in LVb (*Beed et al., 2020*), indicating that indeed LVb neurons might preferentially show persistent activity. Together, these differences will result in differences in information processing.

## Layer Vb neurons preferentially target pyramidal neurons in layers III and II rather than layer II neurons that project to the DG

Both our anatomical and electrophysiological data show that projections from principal neurons in LVb in both entorhinal subdivisions preferentially target pyramidal neurons in LIII and LII. LVb neurons have a weaker relationship with the class of stellate and fan cells in MEC or LEC, respectively. This makes it likely that in both LEC and MEC hippocampal information preferentially interacts with neurons that are part of the LIII-to-CA1/Sub projection system rather than with the LII-to-DG/CA2-3 projecting neurons. Additional target neurons in layer II/III might be the pyramidal neurons that project contralaterally, which in LII belong to the calbindin (CB+) population (*Ohara et al., 2019*; *Steward and Scoville, 1976*; *Varga et al., 2010*), as well as the substantial population of CB+ excitatory intrinsic projection neurons (*Ohara et al., 2019*). The present findings are in line with a previous study using wild-type mice, reporting that most of the inputs to MEC-LII stellate cell arise from superficial layers, whereas those of MEC-LII pyramidal cells arise from the deep layers (*Beed et al., 2010*).

The relatively sparse projection from MEC LVb neurons to LII stellate cells and the more massive projection to LII pyramidal cell was unexpected for two reasons. First, both the stellate and the CB+ population of layer II pyramidal neurons contain grid cells (*Hafting et al., 2005*; *Tang et al., 2014*) and hippocampal excitatory inputs are required for the formation and translocation of grid patterns (*Bonnevie et al., 2013*). Though our data do not exclude that LVb inputs can reach LII stellate cells indirectly through LIII- and LII-pyramidal cells (*Ohara et al., 2019*; *Winterer et al., 2017*), they do indicate that the two populations of grid cells, stellate vs. CB+ cells, might differ with respect to the strength of a main excitatory drive from the hippocampus.

Second, re-entry of hippocampal activity, i.e., the presence of recurrent circuits, has been proposed as one of the mechanisms for temporal storage of information in a neuronal network (*Edelman, 1989*; *Iijima et al., 1996*). Re-entry through LII-to-DG has been observed in in vivo recordings

under anesthesia in rats, although this was examined with current source density analysis, which is not optimal to exclude multisynaptic responses (*Kloosterman et al., 2004*). Such multisynaptic inputs could be mediated by pyramidal neurons in LIII and LII, both of which do contact layer II-to-DG projecting neurons (*Ohara et al., 2019*; *Winterer et al., 2017*). Our current data strongly favor the circuit via LIII-CA1/subiculum in both entorhinal subdivisions to mediate a recurrent hippocampal-entorhinal-hippocampal circuit. The importance of this layer III recurrent network is corroborated by the observation that entorhinal LIII input to the hippocampus field CA1 plays a crucial role in associating temporally discontinuous events and retrieving remote memories (*Lux et al., 2016*; *Suh et al., 2011*).

## The density and strength of the excitatory projection from LVb to LVa appear more prominent in dorsal LEC compared to dorsal MEC

Ever since the seminal observation in monkeys and rats of a hippocampal-cortical projection mediated by layer V of the EC (*Kosel et al., 1982*; *Rosene and Van Hoesen, 1977*), the canonical circuit underlying the hippocampal-cortical interplay, necessary for memory consolidation (*Buzsáki, 1996*; *Eichenbaum et al., 2012*), is believed to use EC LV neurons that receive hippocampal output and send projections to the neocortex. More recent studies in rats and mice indicated that neurons in LVb likely are the main recipients of this hippocampal output stream (*Sürmeli et al., 2015*) and that principal neurons in LVa form the main source of outputs to neocortical areas (*Ohara et al., 2018*; *Sürmeli et al., 2015*). We showed that in LEC as well as ventrally in MEC the LVb-to-LVa connectivity is relatively well developed, in line with our previous study (*Ohara et al., 2018*). In contrast, the relatively sparse connection from LVb-to-LVa in dorsal MEC reported here indicates that at least in dorsal MEC the canonical role of EC LV neurons to mediate hippocampal information transfer to downstream neocortical areas might require a revision. Note that in the present study we focused on the presumed direct excitatory connectivity from LVb-to-LVa neurons using a newly derived TG mouse line. Although we above provided data to argue that we find it unlikely that the reported difference in connectivity between LEC and MEC might be caused by a different preference for a specific cell type in MEC vs. LEC, we cannot completely exclude that option. We further cannot exclude that the apparent differences between dorsal MEC, on the one hand, and ventral MEC and LEC, on the other hand, might be modulated through differences in local circuits, resulting in state changes in the LV network, such as reported in MEC depending on activity in LIII (*Beed et al., 2020*). Likewise, differences in local inhibitory circuits or in the in vivo membrane potential and spike threshold of LVa neurons might be relevant, but in vivo these are unknown. However, our in vitro data do not support the latter possibility. The effectiveness of functional connections may further be influenced by incoming inputs to LVa and/or LVb, such as those from the claustrum (*Kitamura et al., 2017*), medial septum, medial prefrontal cortex, and retrosplenial cortex (*Ohara et al., 2018*). Adding to this complexity is a recent report that there is a direct projection from the intermediate/ventral hippocampus to neurons in MEC LVa (*Rozov et al., 2020*). To assess these complex interactions, further in vivo studies are clearly required.

Finally, it is of interest that LVa, the entorhinal-output layer, is thicker in LEC than in MEC, which might be taken to strengthen our proposal that LEC might be the more relevant player in mediating the hippocampal-cortical interplay relevant for systems memory consolidation (*Buzsáki, 1996*; *Eichenbaum et al., 2012*; *Frankland and Bontempi, 2005*). However, studies that have functionally linked the LVa-output projection with memory consolidation are based on data obtained in MEC (*Kitamura et al., 2017*). In our view, this more likely reflects the strong focus on functions of MEC circuits rather than LEC circuits ever since the discovery of the grid cell (*Hafting et al., 2005*; *Moser et al., 2017*). With the discovery of LEC networks coding for event sequences (*Bellmund et al., 2019*; *Montchal et al., 2019*; *Tsao et al., 2018*), this is likely to change. It is clear that our current suggestion that LEC might be more relevant than MEC in mediating the export of information from the dorsal hippocampus to the neocortex needs to be substantiated in vivo. The functional relevance of the similarities between networks in LEC and MEC mediating re-entry into the hippocampal formation with an apparent preference to target CA1 and subiculum is likewise in need of in vivo studies in order to understand the functional consequences of the present data.

# Materials and methods

**Key resources table**

| Reagent type (species) or resource | Designation | Source or reference | Identifiers | Additional information |
|---|---|---|---|---|
| Strain, strain background (*Mus musculus*) | MEC13-53D | *Blankvoort et al., 2018* | | Not commercially available, but upon request we can send the mouse line. Send email to Dr. Stefan Blankvoort; stefan.blankvoort@ntnu.no. |
| Strain, strain background (*Mus musculus*) | tetO$^{GCaMP6-mCherry}$ | *Blankvoort et al., 2018* | | Not commercially available, but upon request we can send the mouse line. Contact see above. |
| Strain, strain background (*Mus musculus*) | GAD1$^{GFP}$ | *Tamamaki et al., 2003* | | The animals are bred in house after obtaining breeding pairs from Dr Yuchio Yanagawa; yanagawa@med.gunma-u.ac.jp. |
| Strain, strain background (*Mus musculus*) | Gt(ROSA)26Sor$^{tm9(CAG-tdTomato)Hze}$ | The Jackson Laboratory | 007909 RRID:IMSR_JAX:007909 | |
| Genetic reagent (virus) | AAV-TRE-tight-GFP (serotype 2/1) | *Nilssen et al., 2018* | | Viral Vector Core at Kavli Institute for Systems Neuroscience; contact Dr Rajeevkumar Nair Raveendran rajeevkumar.r.nair@ntnu.no. |
| Genetic reagent (virus) | AAV-TRE-tight-oChIEF-citrine (serotype 2/1) | *Nilssen et al., 2018* | | Viral Vector Core at Kavli Institute for Systems Neuroscience; contact see above. |
| Genetic reagent (virus) | AAV-CMV-FLEX-mCherry (serotype 2/1) | This paper | | Viral Vector Core at Kavli Institute for Systems Neuroscience; contact see above. |
| Genetic reagent (virus) | AAVrg-pmSyn1-EBFP-cre | Addgene | 51507 | |
| Genetic reagent (virus) | AAV1.CAG.tdTomato.WPRE.SV40 | Upenn viral core | AV-1-PV3365 | |
| Antibody | Anti-GFP (chicken polyclonal) | Abcam | ab13970 RRID:AB_300798 | (1:500) |
| Antibody | Anti-GFP (rabbit polyclonal) | Thermo Fisher Scientific | A11122 RRID:AB_221569 | (1:2000) |
| Antibody | Anti-PCP4 (rabbit polyclonal) | Sigma Aldrich | HPA005792 RRID:AB_1855086 | (1:300) |
| Antibody | Anti-Ctip2 (rat monoclonal) | Abcam | ab18465 RRID:AB_2064130 | (1:3000) |
| Antibody | Anti-NeuN (guinea pig polyclonal) | Millipore | ABN90P RRID:AB_2341095 | (1:1000) |

*Continued on next page*

*Continued*

| Reagent type (species) or resource | Designation | Source or reference | Identifiers | Additional information |
|---|---|---|---|---|
| Antibody | Anti-NeuN (mouse monoclonal) | Millipore | MAB377 RRID:AB_2298772 | (1:1000) |
| Antibody | Anti-PHA-L (rabbit) | Vector Laboratories | AS-2300 RRID:AB_2313686 | (1:1000) |
| Antibody | Goat anti-chicken IgG (AF 488) | Thermo Fisher Scientific | A11039 RRID:AB_2534096 | (1:400) |
| Antibody | Goat anti-rabbit IgG (AF 546) | Thermo Fisher Scientific | A11010 RRID:AB_2534077 | (1:400) |
| Antibody | Goat anti-rabbit IgG (AF 635) | Thermo Fisher Scientific | A31576 RRID:AB_10374303 | (1:400) |
| Antibody | Goat anti-rat IgG (AF 633) | Thermo Fisher Scientific | A21094 RRID:AB_2535749 | (1:400) |
| Antibody | Goat anti-guinea pig IgG (AF 647) | Thermo Fisher Scientific | A21450 RRID:AB_2735091 | (1:400) |
| Antibody | Goat anti-guinea pig IgG (AF 488) | Thermo Fisher Scientific | A11073 RRID:AB_2534117 | (1:400) |
| Antibody | Goat anti-mouse IgG (AF 488) | Thermo Fisher Scientific | A11001 RRID:AB_2534069 | (1:400) |
| Antibody | Streptavidin, Alexa Fluor 546 conjugate | Thermo Fisher Scientific | S11225 RRID:AB_2532130 | (1:600) |
| Antibody | Cy3 streptavidin | Jackson Immuno Research | 016-160-084 RRID:AB_2337244 | (1:400) |
| Antibody | Neurotrace 640/660 deep-red fluorescent Nissl stain | Thermo Fisher Scientific | N21483 RRID:AB_2572212 | (1:200) |
| Chemical compound, drug | Biotinylated dextran amine | Invitrogen | D1956 | |
| Chemical compound, drug | Phaseolus vulgaris leucoagglutinin | Vector Laboratories | L-1110 | |
| Software, algorithm | Patchmaster | Heka Eletronik | | |
| Software, algorithm | Clampfit | Molecular Devices | | |
| Software, algorithm | MATLAB, 2018a | MathWorks | | |
| Software, algorithm | Image J | http://rsb.info.nih.gov/ij | | |
| Software, algorithm | GraphPad Prism, version 5 | GraphPad software | | |

## Animals

All animals were group housed at a 12:12 hr reversed day/night cycle and had ad libitum access to food and water. Mice of the TG MEC13-53D enhancer strain expressing tTA in PCP4-positive

entorhinal LVb neurons (*Blankvoort et al., 2018*) were used for whole-cell recordings (n = 38) and histological assessment of specific transgene expression (n = 7). To characterize the tTA expression patterns in this mouse line, MEC13-53D was crossed with a tetO$^{GCaMP6-mCherry}$ line (*Blankvoort et al., 2018*; n = 2). Other TG mouse lines, GAD1$^{GFP}$ (*Tamamaki et al., 2003*; n = 4) and Gt(ROSA)26Sor$^{tm9(CAG-tdTomato)Hze}$ (*Madisen et al., 2010*; n = 2), were used to characterize entorhinal neurons in layers Va and Vb. We further used C57BL/6N mice to characterize the morphology of entorhinal LVa neurons (n = 2) and examine the projection of entorhinal LVb neurons and PER neurons in wild-type mice (n = 2). The projection of PER/POR neurons was also examined in MEC13-53D (n = 2). Information on the availability of animals is summarized in the Key Resources Table. All experiments were approved by the local ethics committee and were in accordance with the European Communities Council Directive and the Norwegian Experiments on Animals Act (#17898, #22312).

## Surgical procedures and virus/tracer injections

Animals were anesthetized with isoflurane in an induction chamber (4%, Nycomed, airflow 1 l/min), after which they were moved to a surgical mask on a stereotactic frame (Kopf Instruments). The animals were placed on a heating pad (37°C) to maintain stable body temperature throughout the surgery, and eye ointment was applied to the eyes of the animal to protect the corneas from drying out. The animals were injected subcutaneously with buprenorphine hydrochloride (0.1 mg/kg, Temgesic, Indivior), meloxicam (1 mg/kg, Metacam Boehringer Ingelheim Vetmedica), and bupivacaine hydrochloride (Marcain 1 mg/kg, Astra Zeneca), the latter at the incision site. The head was fixed to the stereotaxic frame with ear bars, and the skin overlying the skull at the incision site was disinfected with ethanol (70%) and iodide before a rostrocaudal incision was made. A craniotomy was made around the approximate coordinate for the injection, and precise measurements were made with the glass capillary used for the virus injection. The coordinates of the injection sites are as follows (anterior to either bregma [APb] or transverse sinus [APt], lateral to sagittal sinus [ML], ventral to dura [DV] in mm): LEC (APt +2.0, ML 3.9, DV 3.0), MEC (APt +1.0, ML 3.3, DV 2.0), nucleus accumbens (NAc) (APb +1.2, ML 1.0, DV 3.8), retrosplenial cortex (RSC) (APb −3.0, ML 0.3, DV 0.8), PER (APb −4.5, ML 4.5, DV 1.5), and POR (APt +1.1, ML 3.3, DV 0.9). Viruses were injected with a nanoliter injector (Nanoliter 2010, World Precision Instruments) controlled by a microsyringe pump controller (Micro4 pump, World Precision Instruments); 100–300 nl of virus was injected with a speed of 25 nl/min. The capillary was left in place for an additional 10 min after the injection, before it was slowly withdrawn from the brain. Finally, the wound was rinsed, and the skin was sutured. The animals were left to recover in a heating chamber, before being returned to their home cage, where their health was checked daily.

For electrophysiological studies, young MEC13-53D mice (5–7 weeks old) were injected with a tTA-dependent AAV (serotype 2/1) carrying either GFP or a fused protein of oChIEF, a variant of the light-activating protein channelrhodopsin2 (*Lin et al., 2009*), and citrine, a yellow fluorescent protein (*Griesbeck et al., 2001*). The construction of these viruses, AAV-TRE-tight-GFP and AAV-TRE-tight-oChIEF-citrine respectively, has been described in *Nilssen et al., 2018*. Data on availability of viral constructs are summarized in the Key Resources Table. These samples were also used to characterize the transgenic mouse line and also the projection patterns of entorhinal LVb neurons. To label LVa neurons, retrograde AAV expressing enhanced blue fluorescent protein (EBFP) and Cre recombinase (AAVrg-pmSyn1-EBFP-cre, Addgene #51507) was injected into either NAc or RSC of Gt(ROSA)26Sor$^{tm9(CAG-tdTomato)Hze}$. LVa neurons were also labeled in C57BL/6N mice by injecting AAVrg-pmSyn1-EBFP-cre in NAc while injecting AAV-CMV-FLEX-mCherry in LEC/MEC. The pAAV-FLEX-mCherry-WPRE construct was created by first cloning a FLEX cassette with MCS into Cla1 and HindIII sites in pAAV-CMV-MCS-WPRE (Agilent) to create pAAV-CMV-FLEX-MCS-WPRE. The sequence of the FLEX cassette was obtained from *Atasoy et al., 2008*. Subsequently, the mCherry sequence was synthesized and cloned in an inverted orientation into EcoR1 and BamH1 sites in pAAV-CMV-FLEX-MCS-WPRE to make pAAV CMV-FLEX-mCherry-WPRE. AAV-CMV-FLEX-mCherry was recovered from pAAV CMV-FLEX-mCherry-WPRE as described elsewhere (*Nair et al., 2020*; *Nilssen et al., 2018*).

For anterograde tracing experiments in wild-type animals, either 2.5% phaseolus vulgaris leucoagglutinin (PHA-L; Vector Laboratories, #L-1110) or 3.5% 10 kDa biotinylated dextran amine (BDA; Invitrogen, #D1956) was injected iontophoretically with positive 6 μA current pulses (6

s on; 6 s off) for 15 min. To label projections from PER in C57BL/6N mice, AAV1.CAG.tdTomato.WPRE.SV40 (Upenn viral core, cat. AV-1-PV3365) was injected iontophoretically with positive 5 μA current pulses (5 s on; 5 s off) for 5 min.

## Acute slice preparation

Two to three weeks after AAV injection, acute slice preparations were prepared as described in detail (*Nilssen et al., 2018*). Briefly, mice were deeply anesthetized with isoflurane and decapitated. The brain was quickly removed and immersed in cold (0°C) oxygenated (95% $O_2$/5% $CO_2$) artificial cerebrospinal fluid (ACSF) containing 110 mM choline chloride, 2.5 mM KCl, 25 mM D-glucose, 25 mM $NaHCO_3$, 11.5 mM sodium ascorbate, 3 mM sodium pyruvate, 1.25 mM $NaH_2PO_4$, 100 mM D-mannitol, 7 mM $MgCl_2$, and 0.5 mM $CaCl_2$, pH 7.4, 430 mOsm. The brain hemispheres were subsequently separated and 350-μm-thick entorhinal slices were cut with a vibrating slicer (Leica VT1000S, Leica Biosystems). We used semicoronal slices for LEC recording, which were cut with an angle of 20° with respect to the coronal plane to optimally preserve neurons and local connections of LEC (*Canto and Witter, 2012b*; *Nilssen et al., 2018*). In case of MEC recordings, horizontal slices were prepared (*Canto and Witter, 2012a*; *Couey et al., 2013*). Slices were incubated in a holding chamber at 35°C in oxygenated ASCF containing 126 mM NaCl, 3 mM KCl, 1.2 mM $Na_2HPO_4$, 10 mM D-glucose, 26 mM $NaHCO_3$, 3 mM $MgCl_2$, and 0.5 mM $CaCl_2$ for 30 min and then kept at room temperature for at least 30 min before use.

## Electrophysiological recording

Patch-clamp recording pipettes (resistance: 4–9 MΩ) were made from borosilicate glass capillaries (1.5 outer diameter × 0.86 inner diameter; Harvard Apparatus) and back-filled with internal solution of the following composition: 120 mM K-gluconate, 10 mM KCl, 10 mM $Na_2$-phosphocreatine, 10 mM HEPES, 4 mM Mg-ATP, 0.3 mM Na-GTP, with pH adjusted to 7.3 and osmolality to 300–305 mOsm. Biocytin (5 mg/ml; Iris Biotech) was added to the internal solution in order to recover cell morphology. Acute slices were moved to the recording setup and visualized using infrared differential interference contrast optics aided by a ×20/1.0 NA water immersion objective (Zeiss Axio Examiner D1, Carl Zeiss). Electrophysiological recordings were performed at 35°C and slices superfused with oxygenated recording ACSF containing 126 mM NaCl, 3 mM KCl, 1.2 mM $Na_2HPO_4$, 10 mM D-glucose, 26 mM $NaHCO_3$, 1.5 mM $MgCl_2$, and 1.6 mM $CaCl_2$. LVb in both LEC and MEC was identified through the presence of the densely packed small cells, and LVb neurons labeled with GFP were selected for recording. LVa neurons were selected for recording on the basis of their large soma size and the fact that they are sparsely distributed directly superficial to the smaller neurons of LVb. Gigaohm resistance seals were acquired for all cells before rupturing the membrane to enter whole-cell mode. Pipette capacitance compensation was performed prior to entering whole-cell configuration, and bridge balance adjustments were carried out at the start of current-clamp recordings. Data acquisition was performed by Patchmaster (Heka Elektronik) controlling an EPC 10 Quadro USB amplifier (Heka Elektronik). Acquired data were low-pass Bessel filtered at 15.34 kHz (for whole-cell current-clamp recording) or 4 kHz (for whole-cell voltage-clamp recording) and digitized at 10 kHz. No correction was made for the liquid junction potential (13 mV as measured experimentally). Data were discarded if the resting membrane potential was $\geq -57$ mV or/and the series resistance was $\geq 40$ MΩ.

Intrinsic membrane properties were measured from membrane voltage responses to step injections of hyperpolarizing and depolarizing current (1 s duration, −200 pA to 200 pA, 20 pA increments). Acquired data were exported to text file with MATLAB (MathWorks) and were analyzed with Clampfit (Molecular Devices). The following electrophysiological parameters analyzed were defined as follows:

> Resting membrane potential ($V_{rest}$; mV): membrane potential measured with no current applied (I = 0 mode).
> Input resistance (Mohm): resistance measured from Ohm's law from the peak of voltage responses to hyperpolarizing current injections (−40 pA injection).
> Time constant (ms): the time it took the voltage deflection to reach 63% of peak of voltage response at hyperpolarizing current injections (−40 pA injection).

Sag ratio (steady-state/peak): measured from voltage responses to hyperpolarizing current injections with peaks at −90 ± 5 mV, as the ratio between the voltage at steady state and the voltage at the peak.

AP threshold (mV): the membrane potential where the rise of the AP was 20 mV/ms.

AP peak (mV): voltage difference from AP threshold to peak.

AP half-width (ms): duration of the AP at half-amplitude from AP threshold.

AP maximum rate of rise (mV/ms): maximal voltage slope during the upstroke of the AP.

Fast AHP (in mV): the peak of AHP in a time window of 6 ms after the membrane potential reached 0 mV during the repolarization phase of AP.

Medium AHP (mV): the peak of AHP in a time window of 200 ms after fast AHP.

AP frequency after 200 pA inj. (Hz): frequency of APs evoked with +200 pA of 1-s-long current injection.

Adaptation: measured from trains of 10 ± 1 APs as [1 -(Last/First)], where Last and First are, respectively, the frequencies of the last and first interspike interval.

DAP: depolarizing voltage deflection after AP. DAP was defined based on previous studies (*Canto and Witter, 2012a*; *Canto and Witter, 2012b*; *Hamam et al., 2000*; *Hamam et al., 2002*).

Optogenetic stimulation and patch-clamp data analysis oChIEF+ fibers were photostimulated with a 473 nm laser controlled by a UGA-42 GEO point scanning system (Rapp OptoElectronic) and delivered through a ×20/1.0 NA WI objective (Carl Zeiss Axio Examiner.D1). Laser pulses had a beam diameter of 36 μm and a duration of 1 ms. The tissue was illuminated with individual pulses at a rate of 1 Hz in a 4 × 5 grid pattern. The grid was positioned such as to allow for the light stimulation to cover across layer I to Va. Laser power was fixed to an intensity that evokes inward currents (EPSCs) but not action potentials. The voltage- or current traces from individual stimulation spots were averaged over 4–6 individual sweeps to create an average response for each point in the 4 × 5 grid. The stimulation point that showed the largest voltage response was used for further analysis. Deflections of the average voltage trace exceeding 10 SDs (±10 SDs) of the baseline were classified as synaptic responses. Postsynaptic potentials were calculated as the difference between the peak of the evoked synaptic potential and the baseline potential measured before stimulus onset. The latency of optical activation was defined as the time interval between light onset and the point where the voltage trace exceeded 10% amplitude. Data analysis was performed using MATLAB. Only slices with at least one successful synaptic response to photostimulation were included in the analysis. There were no outliers that were removed from the data.

All data presented in the figures are shown as mean ± standard errors. Prism software was used for data analysis (GraphPad software), and one-way ANOVA with Bonferroni's multiple comparison test was used to compare the electrophysiological properties and voltage responses between each cell types. To analyze mediolateral gradient in LVb-to-LVa connectivity of MEC, we used Pearson correlation coefficient. A principal component analysis based on the 12 electrophysiological properties (*Figure 2—source data 1*) was conducted in MATLAB. For this purpose, all variables were normalized to a standard deviation of 1.

## Histology, immunohistochemistry, and imaging of electrophysiological slices

After electrophysiological recordings, the brain slices were put in 4% paraformaldehyde (PFA, Merck Chemicals) in 0.1 M phosphate buffer (PB) for 48 hr at 4°C. Slices were permeabilized 5 × 15 min in phosphate buffered saline containing 0.3% Triton X-100 (PBS-Tx) and were immersed in a blocking solution containing PBS-TX and 10% Normal Goat Serum (NGS, Abcam: AB7481) for 3 hr at room temperature. To visualize targeted entorhinal LVb neurons expressing either GFP or oChIEF-citrine, slices were incubated with a primary antibody, chicken anti-GFP (1:500, Abcam, #ab13970), diluted in the blocking solution for 4 days at 4°C. Note that citrine is derived from *Aequorea victoria* GFP (*Griesbeck et al., 2001*), and thus, the signal can be amplified with GFP antibodies. After this, the sections were washed 5 × 15 min in PBS-Tx at room temperature and incubated in a secondary antibody, goat anti-chicken (1:400, Alexa Fluor 488, Thermo Fisher Scientific, #A11039), overnight at room temperature. This secondary antibody incubation was accompanied with the fluorescent conjugated streptavidin (1:600, Alexa Fluor 546, Thermo Fisher Scientific, #S11225) and Neurotrace 640/660 deep-red fluorescent Nissl stain (1:200, Thermo Fisher Scientific, #N21483) in order to stain cells

filled with biocytin and to identify the cytoarchitecture. Slices were rinsed in PBS-TX (3 × 10 min) and dehydrated by increasing ethanol concentrations (30, 50, 70, 90, 100, and 100%, 10 min each). They were treated to a 1:1 mixture of 100% ethanol and methyl salicylate for 10 min before clearing and storage in methyl salicylate (VWR Chemicals).

To image the recorded neurons with a laser scanning confocal microscope (Zeiss LSM 880 Axi-oImager Z2), the slices were mounted in custom-made metal well slides with methyl salicylate and coverslipped. Overview images of the tissue were taken at low magnification (Plan Apochromat ×10, NA 0.45) to confirm the location of the recorded neurons and at higher magnification (Plan Apochromat ×20, NA 0.8) to determine the morphology of the recorded neurons. Both over-view images and high-magnification images were obtained as z-stacks that included the whole extent of each recorded cell to recover the full cell morphology. The morphology of LVb neurons of MEC/LEC was classified based on previous studies (*Canto and Witter, 2012a*; *Canto and Witter, 2012b*; *Hamam et al., 2000*; *Hamam et al., 2002*).

## Histology, immunohistochemistry, and imaging of neuroanatomical tracing samples

After 2–3 weeks of survival, virus- or tracer-injected mice were anesthetized with isoflurane before being euthanized with a lethal intraperitoneal injection of pentobarbital (100 mg/kg, Apotekerfore-ningen). They were subsequently transcardially perfused using a peristaltic pump (World Precision Instruments), first with Ringer's solution (0.85% NaCl, 0.025% KCl, 0.02% NaHCO$_3$) and subsequently with freshly prepared 4% PFA in 0.1 M PB (pH 7.4). The brains were removed from the skull, post-fixed in PFA overnight, and put in a cryo-protective solution containing 20% glycerol, 2% DMSO diluted in 0.125 m PB. A freezing microtome was used to cut the brains into 40-µm-thick sections, which were collected in six equally spaced series for processing.

To enhance the GFP/citrine signal in AAV-TRE-tight-GFP/oChIEF-citrine-infected entorhinal LVb neurons and in GAD67-positive neurons, sections were stained with primary (1:400, chicken anti-GFP, Abcam; 1:2000, rabbit anti-GFP, Thermo Fisher Scientific #A11122) and secondary antibodies (1:400, Alexa Fluor-488 goat anti-chicken IgG, Thermo Fisher Scientific; 1:400, Alexa Fluor-546 goat anti-rabbit IgG, Thermo Fisher Scientific #A11010). This amplification of GFP/citrine signal enabled us to examine the axonal distribution of the infected LVb neurons. Due to the strong neurites label-ing, however, it became difficult to visualize cell bodies of the targeted neurons. For such reasons, we occasionally used the native fluorophore (GFP/citrine) to visualize the cell bodies, while using immunolabeling (anti-GFP/citrine [anti-GFP]) to identify the neurites.

To identify the LVb border and characterize the TG mouse line, LVb neurons were visualized with primary (1:300, rabbit anti-PCP4, Sigma Aldrich #HPA005792; 1:3000, rat anti-Ctip, Abcam #ab18465) and secondary antibodies (1:400, Alexa Fluor 633 goat anti-rat IgG, Thermo Fisher Scien-tific # A21094; 1:400, Alexa Fluor-546 goat anti-rabbit IgG; 1:400, Alexa Fluor 635 goat anti-rabbit IgG, Thermo Fisher Scientific # A31576). For delineation purpose, sections were stained with primary (1:1000, guinea pig anti-NeuN, Millipore #ABN90P; 1:1000, mouse anti-NeuN, Millipore #MAB377) and secondary antibodies (1:400, Alexa Fluor 647 goat anti-guinea pig IgG, Thermo Fisher Scientific #A21450; 1:400, Alexa Fluor 488 goat anti-guinea pig IgG, Thermo Fisher Scientific #A11073; 1:400, Alexa Fluor 488 goat anti-mouse IgG, Thermo Fisher Scientific #A11001). PHA-L was visualized with primary (1:1000, rabbit anti-PHA-L, Vector Laboratories AS-2300) and secondary antibodies (1:400, Alexa Fluor 635 goat anti-rabbit IgG), while BDA was visualized with Cy3-streptavidin (1:400, Jack-son ImmunoResearch 016-160-084).

For all immunohistochemical staining except for Ctip2 staining, we used the same procedure. Sections were rinsed 3 × 10 min in PBS-Tx followed by 60 min incubation in a blocking solution con-taining PBS-Tx with either 5% NGS or 3% bovine serum albumin (BSA). Sections were incubated with the primary antibodies diluted in the blocking solution for 48 hr at 4°C, rinsed 3 × 10 min in PBS-Tx, and incubated with secondary antibodies diluted in PBS-Tx overnight at room temperature. Finally, sections were rinsed 3 × 10 min in PBS. For Ctip2 staining, sections were heated to 80°C for 15 min in 10 mM sodium citrate (SC, pH 8.5). After cooling to room temperature, the sections were permea-bilized by washing them three times in SC buffer containing 0.3% Triton X-100 (SC-Tx) and subse-quently pre-incubated in a blocking solution containing SC-Tx and 3% BSA for 1 hr at room temperature. Next, sections were incubated with the primary antibodies diluted in the blocking solu-tion for 48 hr at 4°C, rinsed 2 × 15 min in SC-Tx and 1% BSA, and incubated with secondary

antibodies diluted in SC-Tx and 1% BSA for overnight at room temperature. Finally, sections were rinsed 3 × 10 min in PBS. After staining, sections were mounted on SuperfrostPlus microscope slides (Thermo Fisher Scientific) in Tris-gelatin solution (0.2% gelatin in Tris-buffer, pH 7.6), dried, and coverslipped with entellan in a toluene solution (Merck Chemicals, Darmstadt, Germany).

Coverslipped samples were imaged using an Axio ScanZ.1 fluorescent scanner, equipped with a ×10 objective, Colibri.2 LED light source, and a quadruple emission filter (Plan Apochromat ×10, NA 0.45, excitation 488/546, emission 405/488/546/633, Carl Zeiss). To quantify the colocalization of GFP, PCP4, and Ctip2 immunolabeling, confocal images were acquired in sections taken at every 240 µm throughout the EC using a confocal microscope (Zeiss LSM 880 AxioImager Z2) with a ×40 oil objective (Plan Apochromat ×40 oil, NA 1.3, Carl Zeiss). The number of immunohistochemically labeled neurons was quantified in a fixed Z-level of the confocal images using Image J software (http://rsb.info.nih.gov/ij).

## Definition of EC and delineations of its layers

The EC of rodents is typically defined as a ventral area of the cerebral cortex, delineated laterally by the rhinal sulcus and medially by the hippocampal formation. It shows a well-established typical laminar structure comprising two superficial and two deep cell layers, separated by a cell-free layer, or lamina dissecans (*Witter, 2011*). Here we divide EC into LEC and MEC, mainly defined based on cytoarchitectural differences in LV, LIII, and LII. MEC has more columnar arrangement of LVb neurons compared to LEC. Layer III in MEC shows a sharp border with the lamina dissecans and has a very homogeneous appearance, whereas in LEC the border is less clear and the cellular organization is much more irregular. Finally, LEC can be identified by the presence of a superficial LII (LIIa) with densely packed neurons that tend to cluster into islands.

The borders of the superficial layers (I–III) and the thin acellular layer IV (lamina dissecans) were delineated as previously described (*Witter, 2011*). Layers Va and Vb were differentiated based on cell size, cell density, cell marker, and the projection patterns. LVb neurons are densely packed small cells that are PCP4-positive, whereas LVa is made up of sparsely distributed large cells, which project to various cortical and subcortical regions (*Figure 1—figure supplement 1*; *Kitamura et al., 2017*; *Ohara et al., 2018*; *Sürmeli et al., 2015*). The border between layer Vb and VI is more difficult to identify since PCP4 staining also labels LVI neurons. We determined this border based on the cell density that decreases in LVI and the overall prominent orientation of neurons parallel to the pial surface (*Witter, 2011*). In case of MEC, the border can also be identified since the typical columnar organization of LVb stops upon entering LVI.

## Statistics

Statistical analyses were performed using GraphPad Prism (GraphPad Software) or MATLAB (MathWorks). The details of tests used are described with the results. Differences between the groups were tested using paired and unpaired t-tests. Group comparisons were made using one-way ANOVA followed by Bonferroni post-hoc tests to control for multiple comparisons. All statistical tests were two-tailed, and thresholds for significance were placed at $*p<0.05$, $**p<0.01$, and $***p<0.001$. All data are shown as mean ± standard errors. No statistical methods were used to predetermine sample size, but the number of mice and cells for each experiment is similar with previous studies in the field (*Doan et al., 2019*; *Nilssen et al., 2018*). Mice were randomly selected from both sexes, and all experiments were successfully replicated in several samples. No blinding was used during data acquisition, but electrophysiological data analyses were performed blind to groups.

## Acknowledgements

We thank Grethe M Olsen and Paulo Girao for help with histological preparations and microscopic imaging, and Paulo Girao and Yasutaka Honda for MATLAB programming. Bente Jacobsen and Thanh Doan contributed the tracing material for *Figure 3—figure supplement 1*, and associated results described.

## Additional information

### Funding

| Funder | Grant reference number | Author |
|---|---|---|
| Kavli Foundation | endowment | Menno P Witter |
| Ministry of Education, Culture, Sports, Science and Technology | KAKENHI (#19K06917) | Shinya Ohara |
| Norwegian Research Council | #197467 | Menno P Witter |
| Norwegian Research Council | #223262 | Menno P Witter |
| Norwegian Research Council | #227769 | Menno P Witter |

The funders had no role in study design, data collection and interpretation, or the decision to submit the work for publication.

### Author contributions

Shinya Ohara, Conceptualization, Data curation, Formal analysis, Investigation, Visualization, Writing - original draft, Project administration, Writing - review and editing; Stefan Blankvoort, Rajeevkumar Raveendran Nair, Clifford Kentros, Resources, Writing - review and editing; Maximiliano J Nigro, Eirik S Nilssen, Conceptualization, Formal analysis, Writing - review and editing; Menno P Witter, Conceptualization, Supervision, Funding acquisition, Writing - original draft, Project administration, Writing - review and editing

### Author ORCIDs

Shinya Ohara (iD) https://orcid.org/0000-0003-0681-5086
Stefan Blankvoort (iD) https://orcid.org/0000-0003-0993-3790
Rajeevkumar Raveendran Nair (iD) https://orcid.org/0000-0001-9755-0874
Maximiliano J Nigro (iD) https://orcid.org/0000-0001-6611-9125
Eirik S Nilssen (iD) http://orcid.org/0000-0001-6997-3343
Clifford Kentros (iD) https://orcid.org/0000-0003-3850-3829
Menno P Witter (iD) https://orcid.org/0000-0003-0285-1637

### Ethics

Animal experimentation: All experiments were approved by the local ethics committee and were in accordance with the European Communities Council Directive and the Norwegian Experiments on Animals Act (#17898, #22312).

### Decision letter and Author response

Decision letter https://doi.org/10.7554/eLife.67262.sa1
Author response https://doi.org/10.7554/eLife.67262.sa2

## Additional files

### Supplementary files

• Transparent reporting form

### Data availability

All data generated or analyzed during this study are included in the manuscript and supporting files. We provide source data for Figure 1, 2, 3, 4, 5, Figure 1-figure supplement 2, and Figure 4-figure supplement 2.

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
