## [Decision Letter]

**Acceptance summary:**

The study addresses a fundamentally important question regarding the connectivity of LVb and LVa in the medial and lateral entorhinal cortex. The authors suggest that the strength of the LVb to LVa connection is stronger in the LEC than in the MEC. This finding will have important implications on studies investigating circuit functions between the hippocampus and the EC. The differential operational principles in the lateral and medial entorhinal cortex is a novel element to the circuit underlying the hippocampal-cortical interplay.

**Decision letter after peer review:**

[Editors’ note: the authors submitted for reconsideration following the decision after peer review. What follows is the decision letter after the first round of review.]

Thank you for submitting your work entitled "Local circuits of layer Vb-to-Va mediate hippocampal-cortical outputs in lateral but not in medial entorhinal cortex" for consideration by *eLife*. Your article has been reviewed by 3 peer reviewers, one of whom is a member of our Board of Reviewing Editors, and the evaluation has been overseen by a Senior Editor. The following individuals involved in review of your submission have agreed to reveal their identity: Matthew F Nolan (Reviewer #2); Desdemona Fricker (Reviewer #3).

Our decision has been reached after consultation between the reviewers. Based on these discussions and the individual reviews below, we regret to inform you that your work will not be considered further for publication in *eLife*.

The study addresses a fundamentally important question regarding the connectivity of LVb and LVa in the medial and lateral entorhinal cortex. The authors suggest that LVb to LVa connection exists in the LEC but not in the MEC. This finding would have important implications on studies investigating circuits functions between the hippocampus and the EC. All three reviewers found the central question important and the data novel. However, there are several technical issues that limit the robustness of the authors claim.

1. While the transgenic animal used in these experiments in elegant and novel, it only labels a subpopulation of the neurons. There is a possibility of selective labelling of neurons with distinct connectivity patterns. The authors would need to show that their approach is not leading to false negative results due to the selective visualization of those neurons that project more modestly to the LVa.

2. The specificity of the injection to the LEC/MEC should be better documented and potential spread to the perirhinal or postrhinal cortex carefully excluded.

3. The findings are presented as LVb to LVa connection did not exist at all in the MEC, however the data shows that the connection is there but it is significantly less dense than in the LEC. Given the graded finding, if the authors aim to show their central claim regarding the lack of mediation of hippocampo-cortical outputs by this connection in the MEC, this would require the addition of functional studies.

Reviewer #1:

The current study by Ohara et al. describes differences in the connectivity patterns between LVb to LVa. The study builds on the authors previous study (Ohara et al. 2018) where they showed the intrinsic connectivity of LVb neurons in the MEC and LEC. The focus of the current study is the difference the authors observed in the strengths of connectivity between LVb and LVa in the MEC and LEC. The authors suggest that the in MEC Vb neurons do not provide substantial direct unput to LVa neurons. The manuscript emphasizes the functional importance of difference as the authors suggest that "……hippocampal -cortex output circuit is present only in LEC, suggesting that episodic systems consolidation predominantly uses LEC-derived information and not allocentric spatial information from MEC."

The study uses a newly developed mouse line to investigate connectivity differences, this is a nice technical approach and the experimental data is of high quality. While the data is solid, the authors tend to overinterpret their findings from the functional point of view. While the observed difference is quite interesting, it is unclear what the impact is on information flow in the MEC and LEC and to which degree they differ functionally. The authors assume major differences and their work is framed based on these expected differences, but the manuscript does not provide data that would demonstrate functionally distinct features.

1. Throughout the text the authors treat their findings as it was 'all-or-none' i.e the LEC has a direct connection between LVb and LVa while the MEC does not. This does not seem to be the case based on their data, the data shows that connectivity in the MEC is less robust but it is definitely there. The difference seems to be quantitative and not qualitative.

2. Due to this problem, the authors seem to be overinterpreting their data by suggesting that the information flow must be significantly different conceptually in the LEC and the MEC and this would have important implications for memory consolidation. It is impossible to draw these conclusions based on the data presented, as there are no experiments investigating the functional, network level consequences of these connectivity differences.

3. The electrophysiology experiments provide information about the basic parameters of the investigated cells, but these lack a physiological context that would allow the authors to evaluate the consequences of these differences on information flow and/or processing in the MEC and the LEC.

4. The study is using a novel transgenic mouse line to differentiate between LVb and LVa neurons, while this is definitely a strength of the study, this strategy allows the authors to visualize ~50% of LEC and ~30% MEC neurons. Since the authors aim to prove a negative (MEC does not have direct connection) the fact that ~70% of the neurons are not labelled could be problematic.

Reviewer #2:

The study investigates key components of the entorhinal circuits through which signals from hippocampus are relayed to the neocortex. The question addressed is important but the stated claim that layer 5b (L5b) to layer 5a (L5a) connections mediate hippocampal-cortical outputs in LEC but not MEC appears to be an over-interpretation of the data. First, the experiments do not test hippocampal to L5a connections, but instead look at L5b to L5a connections. Second, the data provide evidence that there are L5b to L5a projections in LEC and MEC, which contradictions the claim made in the title. These projections do appear denser in LEC under the experimental conditions used, but possible technical explanations for the difference are not carefully addressed. If these technical concerns were addressed, and the conclusions modified appropriately, then I think this study could be very important for the field and would complement well recent work from several labs that collectively suggests that information processing in deep layers of MEC is more complex than has been appreciated (e.g. Sürmeli et al. 2015, Ohara et al. 2018, Wozny et al. 2018, Rozov et al. 2020).

1. An impressive component of the study is the introduction of a new mouse line that labels neurons in layer 5b of MEC and LEC. However, in each area the line appears to label only a subset (30-50%) of the principle cell population. It's unclear whether the unlabelled neurons have similar connectivity to the labelled neurons. If the unlabelled neurons are a distinct sub-population then it's difficult to see how the experiments presented could support the conclusion that L5b does not project to L5a; perhaps there is a projection mediated by the unlabelled neurons? I don't think the authors need include experiments to investigate the unlabelled population, but given that the labelling is incomplete they should be more cautious about generalising from data obtained with the line.

2. For experiments using the AAV conditionally expressing oChIEF-citrine, the extent to which the injections are specific to LEC/MEC is unclear. This is a particular concern for injections into LEC where the possibility that perirhinal or postrhinal cortex are also labelled needs to be carefully considered. For example, in Figure 3D it appears the virus has spread to perirhinal cortex. If this is the case then axonal projections/responses could originate there rather than from L5b of LEC. I suggest to exclude any experiments were there is any suggestion of expression outside LEC/MEC or where this cannot be ruled out through verification of the labelling. Alternatively, one might include control experiments in which the AAV is targeted to perirhinal and postrhinal cortex. Similar concerns should be addressed for injections that target the MEC to rule out spread to the pre/parasubiculum.

3. It appears likely from the biocytin fills shown that the apical dendrites of some of the recorded L5a neurons have been cut (e.g. Figure 4A, Figure 4-Supplement 1D, neuron v). Where the apical dendrite is clearly intact and undamaged synaptic responses to activation of L5b neurons are quite clear (e.g. Figure 4-Supplement 1D, neuron x). Given that axons of L5b cells branch extensively in L3, it is possible that any synapses they make with L5a neurons would be on their apical dendrites within L3. It therefore seems important to restrict the analysis only to L5a neurons with intact apical dendrites; a reasonable criteria would be that the dendrite extends through L3 at a reasonable distance (> 30 μm?) below the surface of the slice.

4. Throughout the manuscript the data is over-interpreted. Here are some examples:

– The title over-extrapolates from the results and should be changed. A more accurate title would be along the lines of "Evidence that L5b to L5a connections are more effective in lateral compared to medial entorhinal cortex".

– "the conclusion that the dorsal parts of MEC lacks the canonical hippocampal-cortical output system" seems over-stated given the evidence (see comments above).

– Discussion, para 1, "Our key finding is that LEC and MEC are strikingly different with respect to the hippocampal-cortical pathway mediated by LV neurons, in that we obtained electrophysiological evidence for the presence of this postulated crucial circuit in LEC, but not in MEC". This is misleading as there is also evidence for L5b to L5a connections in MEC, although this projection may be relatively weak. Recent work by Rozov et al. demonstrating a projection from intermediate hippocampus to L5a provides good evidence for an alternative model in which MEC does relay hippocampal outputs. This needs to be considered.

5. What proportion of responses are mono-synaptic? How was this tested?

Reviewer #3:

The present manuscript focuses on a subpopulations of layer 5 neurons in medial and lateral entorhinal cortex and its functional connections to target neurons in layers 2, 3 and 5. The authors show a difference in LVb-to-LVa connectivity between MEC and LEC. The results suggest that the entorhinal output circuit via LVb-to-LVa is present primarily in LEC.

The work relies on and is made possible by a newly described transgenic mouse (TG) where LVb neurons can be labeled and stimulated with light. The authors showed that these neurons are largely co-labeled with PCP4, a marker for LVb. They compared the apical dendritic extent from TG labeled cells (LVb) and Nac retrogradely labeled cells (LVa) in medial and lateral EC. The intrinsic electrophysiological properties of LVa and LVb neurons were measured and used for PCA showing segregation according to sublayer and region. The axonal distribution and translaminar local connections of LVb neurons form the TG mice were then examined. Cells were recorded in vitroin vitro and filled with biocytin, both from MEC and LEC, with multiple cells in a same slice, documented with exquisite high quality images. The study of the LVb translaminar connectivity via a direct comparison of postsynaptic responses in neurons in different layers in a same slice is gold standard for this type of functional connectivity analysis. There is also an investigation of mixed excitatory-inhibitory postsynaptic response sequences, and evidence for a dorso-ventral gradient in LVb-to-LVa connectivity in MEC is given.

The study combines TG mice, immunolabeling, retrograde labeling, morphological analysis and in vitro electrophysiology with optogenetic photo-stimulation. While it builds on already published work by the same group and others, by comparing the local target neurons of LVb in MEC and in LEC, the manuscript provides a unique contribution to the literature on the laminar circuit organization in the Entorhinal Cortex. In view of the central position of this area in the hippocampal memory systems of the rodent brain, these results are of interest to a broader neuroscience audience. It is also a nice example of a bottom-up approach, where data on the entorhinal translaminar connectivity may influence and constrain theories of hippocampal-cortical processing.

1. Almost all TG labelled neurons are positive for PCP4 but not so vice versa, only 45.9 and 30.P% of PCP4 + neurons in LEC and MEC are labeled in the TG mouse (page 5) leaving open the possibility that the TG mous labels a (specific?) subset of LVb neurons. Did you test whether TG labeled LVb cells co-localize with Ctip2 ?

2. The direct comparison of translaminar connectivity of LVb neurons is very convincing. But if your main conclusion (title) concerns the difference of LVb-to-LVa connectivity between MEC and LEC, it would have been more appropriate to test that in a same slice. I think this point should not stand in the way of publication, but while the data strongly support conclusions on the laminar differences of LVb connectivity, the evidence for differences in LVb-to-LVa connectivity between MEC and LEC is a bit weaker and more indirect.

3. Postsynaptic responses (in mV) in LEC are about twice as high in amplitude as in MEC (Figure 4E vs Figure 5E), across all layers. Please discuss possible reasons, and possible impact on the circuit function. Is the probability to initiate action potentials higher in LEC ?

4. Give the onset latencies of postsynaptic excitatory potentials induced by LVb photostimulation. Are latencies monosynaptic? Or also polysynaptic? Ideally this could be tested by applying a cocktail of TTX-4-AP.

5. Figure 4 S3, Figure 5 S2. Analysis of inhibition. What is the cut-off criterium to say inhibition is present or not? It might be more appropriate to give the I/E ratio.

[Editors’ note: further revisions were suggested prior to acceptance, as described below.]

Thank you for resubmitting your work entitled "Local projections of layer Vb-to-Va are more effective in lateral than in medial entorhinal cortex" for further consideration by *eLife*. Your revised article has been reviewed by three reviewers, one of whom is a member of our Board of Reviewing Editors, and the evaluation has been overseen by Laura Colgin as the Senior Editor.

The study addresses a fundamentally important question regarding the connectivity of LVb and LVa in the medial and lateral entorhinal cortex. The authors suggest that the strength of the LVb to LVa connection is stronger in the LEC than in the MEC. This finding would have important implications on studies investigating circuits functions between the hippocampus and the EC. All three reviewers found the central question important and the data novel. However, the text overstates the functional implications of the findings and an important alternative explanation regarding potential differences in cut dendrites haven't been fully excluded.

The manuscript has been improved but there are some remaining issues that need to be addressed, as outlined below:

There are two major issues that need to be addressed in the revised in the manuscript.

1. The current version still overstates the differences in the effectiveness of the connections; the functional data is not robust enough to draw these conclusions and the claims should be restated.

2. Previously raised issue regarding the potential impact of cut dendrites is still not resolved. The authors would need to provide more in depth analysis to address/exclude this possibility fully.

Other, smaller issues listed in the reviewers' reports that also would need to take into consideration during the revision.

Reviewer #1:

The authors have improved the manuscript based on previous comments. However, I still think that the functional aspect and implications are overstated in the revised manuscript. The physiological data demonstrate the connectivity's strengths (and hence supports the anatomy data), but this alone won't be sufficient to comment on the functional effectiveness of the connections, which would also depend on other factors that are not investigated here.

Reviewer #2:

This study presents data about the connectivity from subpopulations of neurons in layer 5b of the medial and lateral entorhinal cortices to neurons in more superficial layers. This data is likely to make an important contribution to understanding the circuit basis for memory and spatial cognition.

The major weakness of the study is that the claim that L5b -> L5a connections are more effective in the LEC than MEC continues to be over-stated given the evidence. There is a risk here that this will have an impact, but beyond what the data actually establish and possibly in a way that will hinder rather than advance the field. In my opinion, the more specific claim, that differences in L5b -> L5a connectivity suggest different processing mechanisms with as yet unknown functional impact would be sufficiently important to justify publication in *eLife*.

A remaining technical weakness is that the possibility of cut dendrites explaining differences in synaptic responses is not addressed sufficiently convincingly.

It should be possible to address these issues without additional experimentation.

The strengths of the manuscript are the combined anatomical and electrophysiological approach that leverages a novel mouse line, and the likely importance of the observations to investigation of circuit mechanism for spatial cognition and memory.

While the manuscript is greatly improved there are remaining weaknesses.

1. While the over-interpretation of the data is reduced compared to the previous version of the manuscript, this still substantially compromises the study. The manuscript continues to make claims about functional effectiveness of connections that are not sufficiently supported by the data. A more specific claim that glutamateric connections from L5b to L5a appear denser in MEC than LEC, and that this suggests different operating principles for computation in these two regions, would be well justified by the data, would be of wide interest, would do much to stimulate future work and would not lead to erroneous interpretations down the line. Specific statements that over-interpret the data should be amended.

2. The Discussion could do a much better job of considering caveats in the interpretation of the data. In particular, relatively greater excitatory connectivity in a brain slice does not necessarily imply more effective functional connectivity. Alternative scenarios to consider could include the following. 1. It's unclear from the present data whether the relative difference in excitatory input from L5b to L5a is matched by a difference in feedforward or feedback inhibition. If it is then the pathways may turn out to be similarly effective. Testing this convincingly will require in vivo experiments. 2. Either pathway could be subject to neuromodulation in vivo. It's conceivable this could substantially modify or even reverse their apparent relative effectiveness. 3. The membrane potential and spike threshold of L5a neurons in vivo is unknown. If the membrane potential of L5a neurons in MEC is more depolarized in vivo than in LEC, then the L5b to L5a pathway could turn out to be more effective. The discussion should also consider the fact that the mouse line used labels only a sub-set of the neurons in L5b of LEC and MEC. The possibility that connectivity from other L5b neurons may differ should be clearly noted.

3. The possibility that differences in cut dendrites explain the differences in synaptic responses between LEC and MEC, and within the MEC, is still not ruled out. The revised manuscript now gives an indication of the number of dendritic branches in MEC that are > 150 μm. This is insufficient as layer 3 can be on the order of 300 μm or more in width between its superficial and deep borders, while the axons from L5b extend well into L2. To make the claims convincing it's important that adequate quantification of remaining apical dendritic length (and if possible diameter/surface areas) in LEC and at each dorsoventral level of MEC is included (for both cut and not-cut neurons).

4. I couldn't find anywhere whether any connections were tested with glutamate receptor antagonists to confirm they are synaptic and glutamatergic. It's unlikely but not impossible that some short latency responses reflect low level expression of ChR2 in non-5b neurons (in my experience this might not be detectable from fluorescence). It would be good to know this can be ruled out.

Reviewer #3:

I have read the authors' responses and I feel that they are adequate. I have no further comments.

---

## [Author Response]

[Editors’ note: the authors resubmitted a revised version of the paper for consideration. What follows is the authors’ response to the first round of review.]

[…] All three reviewers found the central question important and the data novel. However, there are several technical issues that limit the robustness of the authors claim.1. While the transgenic animal used in these experiments in elegant and novel, it only labels a subpopulation of the neurons. There is a possibility of selective labelling of neurons with distinct connectivity patterns. The authors would need to show that their approach is not leading to false negative results due to the selective visualization of those neurons that project more modestly to the LVa.

This is certainly a correct concern, since it is well known that layer Vb in rodents comprises multiple neuron types, both in LEC and MEC. In the initial version, we tried to address this concern in the discussion, by mentioning that traditional anterograde tracing methods result in similar patterns. We indeed did not provide additional experimental data to substantiate that this is not a concern. We have now added experimental data on the sparsity of projections from the general population of neurons in LVb of MEC to LVa, in both rats and mice (Figure 3—figure supplement 1. D, E). We further maintained referencing existing literature in the rat (Köhler, 1986) and in the present version added a reference to a study in the monkey (Chrobak and Amaral, 2007), both substantiating the apparent differences between LEC and MEC regarding Vb to Va projections in wildtype animals using anterograde tracing.

2. The specificity of the injection to the LEC/MEC should be better documented and potential spread to the perirhinal or postrhinal cortex carefully excluded.

This is a fair criticism, which we originally assumed was covered sufficiently in the discussion since it is known that deep layers of PER and POR in the rat do not substantially contribute to projections to LEC and MEC (Burwell and Amaral, 1998). However, since we do have experimental data in both wildtype and the TG mice, we have added this as a supplementary figure to the manuscript (Figure 3—figure supplement 1A—C). In none of these experiments did we notice substantial projections to neither LEC or MEC, so the very sparse spill of the AAV infection into PER/POR in our experimental cases with injections in LVb of MEC or LEC is unlikely to have affected our results.

3. The findings are presented as LVb to LVa connection did not exist at all in the MEC, however the data shows that the connection is there but it is significantly less dense than in the LEC. Given the graded finding, if the authors aim to show their central claim regarding the lack of mediation of hippocampo-cortical outputs by this connection in the MEC, this would require the addition of functional studies.

We do agree that we have emphasized and simplified our findings too much, using inappropriate wording. We will respond to this in more detail below, but want to add here that our data now only show that the projection is less dense (and indeed significantly less as our new, added analyses show), but also that in MEC the projection shows less synaptic strength.

Reviewer #1:[…] The study uses a newly developed mouse line to investigate connectivity differences, this is a nice technical approach and the experimental data is of high quality. While the data is solid, the authors tend to overinterpret their findings from the functional point of view. While the observed difference is quite interesting, it is unclear what the impact is on information flow in the MEC and LEC and to which degree they differ functionally. The authors assume major differences and their work is framed based on these expected differences, but the manuscript does not provide data that would demonstrate functionally distinct features.

We appreciate that this reviewer, as do the other reviewers, indicate that we have used incorrect wordings, leaving the reader with the impression that there is no projection from LVb to LVa in dorsal MEC at all, and that we provide data supporting functional consequences. We have carefully rephrased this throughout the manuscript.

1. Throughout the text the authors treat their findings as it was 'all-or-none' i.e the LEC has a direct connection between LVb and LVa while the MEC does not. This does not seem to be the case based on their data, the data shows that connectivity in the MEC is less robust but it is definitely there. The difference seems to be quantitative and not qualitative.

We have carefully corrected our wording and now also emphasize that the lesser connectivity is much more prominent in dorsal MEC and less so in ventral MEC. Although the data were already in the original manuscript (Figure 4—figure supplement 2.), we ignored this in how we phrased our final conclusions (see also our responses below).

2. Due to this problem, the authors seem to be overinterpreting their data by suggesting that the information flow must be significantly different conceptually in the LEC and the MEC and this would have important implications for memory consolidation. It is impossible to draw these conclusions based on the data presented, as there are no experiments investigating the functional, network level consequences of these connectivity differences.

We appreciate this comment and as correctly mentioned by the reviewer, we indeed suggest, but not conclude, that functional differences of LEC and MEC regarding memory consolidation may exist. We do however realize that we seem to conclude that such is the case which is incorrect indeed. We carefully corrected our wordings throughout so that it would not lead to any misunderstanding.

3. The electrophysiology experiments provide information about the basic parameters of the investigated cells, but these lack a physiological context that would allow the authors to evaluate the consequences of these differences on information flow and/or processing in the MEC and the LEC.

We most certainly agree with the comment that our data do not allow us to evaluate the consequences of the reported differences on information flow and we refrain from aiming to do so. We merely suggest that our data point to potentially interesting differences that need to be explored. It would take quite some time to obtain the results requested by the reviewer, which is beyond what we feel feasible at this point in time. In the original manuscript, we already included some functional comments related to main differences in physiological properties of neuron types in layers II and III. In the revised manuscript, we added some comments concerning potential functional consequences of the reported differences in neurons in LVa and LVb and between LVb neurons in LEC versus MEC. However, since firing patterns of neurons result from the complex interactions of morphological structure, the types and somatodendritic distribution of voltage-gated channels and many other factors, we refrain from detailed functional speculations since these are not related to the core message we aim to convey with this paper.

4. The study is using a novel transgenic mouse line to differentiate between LVb and LVa neurons, while this is definitely a strength of the study, this strategy allows the authors to visualize ~50% of LEC and ~30% MEC neurons. Since the authors aim to prove a negative (MEC does not have direct connection) the fact that ~70% of the neurons are not labelled could be problematic.

As indicated above (response to general comment 1), we added a supplementary figure which shows the projection of layer Vb neurons in MEC of wild-type mice, which was also confirmed in wild-type rats (Figure 3—figure supplement 1). These additional data show that MEC LVb neurons project more heavily to LIII than to LVa, thus indicating that our results are not due to the labeling of only a subpopulation in MEC-LVb. The sparse MEC LVb-to-LVa connectivity has also been reported in a

recent paper (Rozov et al., 2020), which we now reference in our discussion. Note that also the PCP4 data, representing a much larger population of LVb neurons, show a similar lack of axons in LVa in MEC (Figure 1—figure supplement 1). We have emphasized this is the revised version.

Reviewer #2:The study investigates key components of the entorhinal circuits through which signals from hippocampus are relayed to the neocortex. The question addressed is important but the stated claim that layer 5b (L5b) to layer 5a (L5a) connections mediate hippocampal-cortical outputs in LEC but not MEC appears to be an over-interpretation of the data. First, the experiments do not test hippocampal to L5a connections, but instead look at L5b to L5a connections. Second, the data provide evidence that there are L5b to L5a projections in LEC and MEC, which contradictions the claim made in the title. These projections do appear denser in LEC under the experimental conditions used, but possible technical explanations for the difference are not carefully addressed. If these technical concerns were addressed, and the conclusions modified appropriately, then I think this study could be very important for the field and would complement well recent work from several labs that collectively suggests that information processing in deep layers of MEC is more complex than has been appreciated (e.g. Sürmeli et al. 2015, Ohara et al. 2018, Wozny et al. 2018, Rozov et al. 2020).1. An impressive component of the study is the introduction of a new mouse line that labels neurons in layer 5b of MEC and LEC. However, in each area the line appears to label only a subset (30-50%) of the principle cell population. It's unclear whether the unlabelled neurons have similar connectivity to the labelled neurons. If the unlabelled neurons are a distinct sub-population then it's difficult to see how the experiments presented could support the conclusion that L5b does not project to L5a; perhaps there is a projection mediated by the unlabelled neurons? I don't think the authors need include experiments to investigate the unlabelled population, but given that the labelling is incomplete they should be more cautious about generalising from data obtained with the line.

See our response to comment 4 of reviewer 1.

2. For experiments using the AAV conditionally expressing oChIEF-citrine, the extent to which the injections are specific to LEC/MEC is unclear. This is a particular concern for injections into LEC where the possibility that perirhinal or postrhinal cortex are also labelled needs to be carefully considered. For example, in Figure 3D it appears the virus has spread to perirhinal cortex. If this is the case then axonal projections/responses could originate there rather than from L5b of LEC. I suggest to exclude any experiments were there is any suggestion of expression outside LEC/MEC or where this cannot be ruled out through verification of the labelling. Alternatively, one might include control experiments in which the AAV is targeted to perirhinal and postrhinal cortex. Similar concerns should be addressed for injections that target the MEC to rule out spread to the pre/parasubiculum.

See our response above to the general comment 2. We have added supplementary data with AAV injection in deep PER (Figure 3—figure supplement 1A, B) in TG-animals as well as anterograde tracer injections in wildtype mice. We hardly observed labeled fibers in LEC originating from deep PER, which indicate that our data collected in LEC were not affected by the sparse neurons in deep PER labeled by our AAV-injections. We included similar experimental data for POR (Figure 3—figure supplement 1C). Regarding spread of AAV from MEC into pre/parasubiculum, we feel that this is not an issue, since we report a diminished projection to LVa, not an added component. Moreover, it is very well established that pre/parasubiculum preferentially if not exclusively project to superficial layers. However, the reviewer is correct that these inputs might target neurons in LVa, as we have shown in the rat (Canto et al., 2012). So, this might have resulted in an overestimation of the inputs to LVa neurons, but even so, the input synaptic strength and proportion of responsive neurons never reaches the level as seen in LVa of LEC. Note that we did stimulate the slices throughout all layers, and we have emphasized this in the method section of the paper (see also our reply to comment 3).

3. It appears likely from the biocytin fills shown that the apical dendrites of some of the recorded L5a neurons have been cut (e.g. Figure 4A, Figure 4-Supplement 1D, neuron v). Where the apical dendrite is clearly intact and undamaged synaptic responses to activation of L5b neurons are quite clear (e.g. Figure 4-Supplement 1D, neuron x). Given that axons of L5b cells branch extensively in L3, it is possible that any synapses they make with L5a neurons would be on their apical dendrites within L3. It therefore seems important to restrict the analysis only to L5a neurons with intact apical dendrites; a reasonable criteria would be that the dendrite extends through L3 at a reasonable distance (> 30 μm?) below the surface of the slice.

As correctly pointed out by the reviewer, all L5a neurons which respond to the light stimulation indeed have intact apical dendrites in L3. However, we also did observe many neurons with apical dendrites (> 150 μm in L3) which do not respond in dorsal- (N=3) and middle-level slices (N=6). These data indicate that the differences of LVb-to-LVa connectivity along the dorso-ventral axis were not due to the severing of apical dendrites. We added the information of severed apical dendrites of L5b neurons in “Figure 4—figure supplement 2”. Note that in all experiments we systematically optically stimulate in all layers of the slice allowing us to assess the possible impact of axons in layers III and II.

4. Throughout the manuscript the data is over-interpreted. Here are some examples:– The title over-extrapolates from the results and should be changed. A more accurate title would be along the lines of "Evidence that L5b to L5a connections are more effective in lateral compared to medial entorhinal cortex".

We thank the reviewer for this suggestion and have changed our title accordingly.

– "the conclusion that the dorsal parts of MEC lacks the canonical hippocampal-cortical output system" seems over-stated given the evidence (see comments above).

We have rephrased the sentence and carefully checked the manuscript to correct any such overstatements.

– Discussion, para 1, "Our key finding is that LEC and MEC are strikingly different with respect to the hippocampal-cortical pathway mediated by LV neurons, in that we obtained electrophysiological evidence for the presence of this postulated crucial circuit in LEC, but not in MEC". This is misleading as there is also evidence for L5b to L5a connections in MEC, although this projection may be relatively weak. Recent work by Rozov et al. demonstrating a projection from intermediate hippocampus to L5a provides good evidence for an alternative model in which MEC does relay hippocampal outputs. This needs to be considered.

Thank you for suggesting this paper, but please note that this paper was published after we submitted the paper to *eLife*. We now included this paper in our discussion, and carefully stated that our suggestion concerning a weaker output hippocampal-cortical circuit is the case in the dorsal hippocampal-medial entorhinal circuit in the text and also in the conclusion figure (Figure 6).

5. What proportion of responses are mono-synaptic? How was this tested?

This is an interesting question, but we did not test this directly, using standard pharmacological approaches. We have added information on latency of the measured responses in Figures 4 and 5 (see also the request of Reviewer 3). In a series of experiments carried out in parallel in mouse LEC LII, using similar viral and laser stimulation protocols, we measured latencies ranging from 4.5 – 6.8 msec, which consistently were shown to be monosynaptic when tested using pharmacological measures (applying TTX and 4-AP to the bath; E.S. Nilssen, 2019 Synaptic connectivity of principal cells in layer II of the lateral entorhinal cortex. PhD Thesis NTNU, ISSN 1503-8181). Based on the present latency data, we find it likely that most recorded responses are monosynaptic, though we cannot exclude that some of the connections, particularly the ones from MEC LVb to LVa neurons and to LII stellate cells include or represent disynaptic responses. We have inserted a short note on this in the manuscript. When we planned these experiments, we did not deem pharmacological checks on monosynaptic contacts all that relevant; even if there are disynaptic or polysynaptic responses, the fact that the overall number of responding cells and the synaptic strengths is lower in LVa in dorsal MEC compared to ventral MEC and LEC stands.

Reviewer #3:[…] 1. Almost all TG labelled neurons are positive for PCP4 but not so vice versa, only 45.9 and 30.P% of PCP4 + neurons in LEC and MEC are labeled in the TG mouse (page 5) leaving open the possibility that the TG mous labels a (specific?) subset of LVb neurons. Did you test whether TG labeled LVb cells co-localize with Ctip2 ?

We haven’t checked whether TG labeled LVb neurons co-localize with Ctip2. However, as shown in “Figure 1—figure supplement 2”, we confirmed that Ctip2-positive neurons incorporate PCP4positive neurons, and in addition, labels also PCP4-negative neurons (yellow and white arrowheads). Therefore, it is likely that the TG labeled LVb neurons co-localize with Ctip2.

2. The direct comparison of translaminar connectivity of LVb neurons is very convincing. But if your main conclusion (title) concerns the difference of LVb-to-LVa connectivity between MEC and LEC, it would have been more appropriate to test that in a same slice. I think this point should not stand in the way of publication, but while the data strongly support conclusions on the laminar differences of LVb connectivity, the evidence for differences in LVb-to-LVa connectivity between MEC and LEC is a bit weaker and more indirect.

We are not entirely sure what the reviewer is getting at. If the point is that the difference between LEC and MEC is not tested in the same slice, that is a correct statement, but to do that is technically unrealistic. In line to our replies to the previous reviewers, the main point is that in dorsal MEC the projections from LVb to LVa are very different from what we observe in LEC and ventral MEC.

However, to get dorsal MEC and LEC in the same slice is impossible and although dorsal and ventral MEC might be captured in one sagittal section, one need to inject AAV at multiple dorsoventral levels to do such recordings. The risk of having too much spread of virus in such cases would be too high and we do not find it ethically acceptable to use many animals in the hope of getting one (at best) successful animal. The number of animals used and the number of neurons we recorded from provide a reliable statistical sample to conclude that there are differences in the local circuits.

3. Postsynaptic responses (in mV) in LEC are about twice as high in amplitude as in MEC (Figure 4E vs Figure 5E), across all layers. Please discuss possible reasons, and possible impact on the circuit function. Is the probability to initiate action potentials higher in LEC ?

We thank the reviewer for this suggestion, and we agree that we should have discussed this. We don’t have any data that indicate that the probability to initiate action potentials higher in LEC than in MEC and this is in line with data in two published studies in the rat where elementary biophysical properties were systematically compared between neurons in all layers of LEC and MEC in postnatal rats (Canto et al. 2012a and b) As discussed at some length in Canto and Witter (2012a) most studies carried out in adult rodents are in line with these observations. We deem it most likely that the differences in the generated postsynpatic amplitudes are due to the differences in the number of oChiEF-expressing neurons and thus the resulting differenced in labeled axonal densities. As shown in Figure 1E, the proportion of tTA-expressing neurons is higher in LEC than in MEC. We have inserted a comment along these lines in the revised manuscript.

4. Give the onset latencies of postsynaptic excitatory potentials induced by LVb photostimulation. Are latencies monosynaptic? Or also polysynaptic? Ideally this could be tested by applying a cocktail of TTX-4-AP.

Please see our reply to major comment 5 of reviewer 2 above.

5. Figure 4 S3, Figure 5 S2. Analysis of inhibition. What is the cut-off criterium to say inhibition is present or not? It might be more appropriate to give the I/E ratio.

We agree that our analysis on the inhibition may not be appropriate. In view of the very low number of observations we do not have sufficient data to provide proper I/E ratios, and since the presence or absence of inhibition does not affect our conclusion, we removed the data related to the inhibition.

[Editors’ note: what follows is the authors’ response to the second round of review.]

[…] All three reviewers found the central question important and the data novel. However, the text overstates the functional implications of the findings and an important alternative explanation regarding potential differences in cut dendrites haven't been fully excluded.The manuscript has been improved but there are some remaining issues that need to be addressed, as outlined below:There are two major issues that need to be addressed in the revised in the manuscript.1. The current version still overstates the differences in the effectiveness of the connections; the functional data is not robust enough to draw these conclusions and the claims should be restated.2. Previously raised issue regarding the potential impact of cut dendrites is still not resolved. The authors would need to provide more in depth analysis to address/exclude this possibility fully.Other, smaller issues listed in the reviewers' reports that also would need to take into consideration during the revision.

We are very happy with the overall positive evaluation of the revised version. Both reviewer 1 and 2 indicate clearly that although the data as such are very relevant for the field and merit publication, we are overstating the functional consequences of our findings. Though we do appreciate this evaluation and have revised the paper accordingly, we want to emphasize that during the preparation of the first version as well as the revision we have consulted some experts/theoreticians in the field of memory consolidation who convinced us about the potential importance of our findings related to memory consolidation and our choice for wording has been strongly influenced by these detailed and stimulative discussions.

We have thoroughly addressed all issues as indicated in detail below.

Reviewer #1:The authors have improved the manuscript based on previous comments. However, I still think that the functional aspect and implications are overstated in the revised manuscript. The physiological data demonstrate the connectivity's strengths (and hence supports the anatomy data), but this alone won't be sufficient to comment on the functional effectiveness of the connections, which would also depend on other factors that are not investigated here.

We thank the reviewer for the overall positive evaluation of the revised version. We fully agree with the reviewer that we do not provide experimental data allowing us to conclude on the functional effectiveness of the connections. However, we respectfully disagree that our data would not allow us to comment on this aspect or point to potential relevant functional consequences. We are aware of the fact that many other factors will impact these functional consequences in vivo and have elaborated on that a bit more in the discussion (pages 18/19, lines 460-472 of the revised manuscript). We have scrutinized all comments on functional effectiveness and reworded them to the best of our skills, particularly guided by the detailed suggestions of reviewer 2 (see below).

Reviewer #2:This study presents data about the connectivity from subpopulations of neurons in layer 5b of the medial and lateral entorhinal cortices to neurons in more superficial layers. This data is likely to make an important contribution to understanding the circuit basis for memory and spatial cognition.The major weakness of the study is that the claim that L5b -> L5a connections are more effective in the LEC than MEC continues to be over-stated given the evidence. There is a risk here that this will have an impact, but beyond what the data actually establish and possibly in a way that will hinder rather than advance the field. In my opinion, the more specific claim, that differences in L5b -> L5a connectivity suggest different processing mechanisms with as yet unknown functional impact would be sufficiently important to justify publication in eLife.A remaining technical weakness is that the possibility of cut dendrites explaining differences in synaptic responses is not addressed sufficiently convincingly.It should be possible to address these issues without additional experimentation.

We thank the reviewer for supporting the publication of our data and manuscript in *eLife* and we are grateful for the detailed level of comments and suggestions to reword our manuscript to improve how to accurately convey the data and the conclusions, as well as how we evaluate and interpret our data.

The strengths of the manuscript are the combined anatomical and electrophysiological approach that leverages a novel mouse line, and the likely importance of the observations to investigation of circuit mechanism for spatial cognition and memory.While the manuscript is greatly improved there are remaining weaknesses.1. While the over-interpretation of the data is reduced compared to the previous version of the manuscript, this still substantially compromises the study. The manuscript continues to make claims about functional effectiveness of connections that are not sufficiently supported by the data. A more specific claim that glutamateric connections from L5b to L5a appear denser in MEC than LEC, and that this suggests different operating principles for computation in these two regions, would be well justified by the data, would be of wide interest, would do much to stimulate future work and would not lead to erroneous interpretations down the line. Specific statements that over-interpret the data should be amended.

We thank the reviewer for this constructive remark, and we have reformulated our main conclusions and suggestions throughout accordingly and tried to clearly differentiate between conclusions and suggestions. The relevant changes include sentences in the abstract, the introduction and the discussion mainly. They can be easily spotted in the version in which we indicated the main changes we made in the revised version (see also our comment to point two).

2. The Discussion could do a much better job of considering caveats in the interpretation of the data. In particular, relatively greater excitatory connectivity in a brain slice does not necessarily imply more effective functional connectivity. Alternative scenarios to consider could include the following. 1. It's unclear from the present data whether the relative difference in excitatory input from L5b to L5a is matched by a difference in feedforward or feedback inhibition. If it is then the pathways may turn out to be similarly effective. Testing this convincingly will require in vivo experiments. 2. Either pathway could be subject to neuromodulation in vivo. It's conceivable this could substantially modify or even reverse their apparent relative effectiveness. 3. The membrane potential and spike threshold of L5a neurons in vivo is unknown. If the membrane potential of L5a neurons in MEC is more depolarized in vivo than in LEC, then the L5b to L5a pathway could turn out to be more effective. The discussion should also consider the fact that the mouse line used labels only a sub-set of the neurons in L5b of LEC and MEC. The possibility that connectivity from other L5b neurons may differ should be clearly noted.

We appreciate the suggestions of the reviewer and we fully agree that all the suggested, and even more scenarios might be relevant. To cover them all would go far beyond the scope of the paper or would simply result in a listing of possibilities with references. Of course, we cannot exclude any of the ones suggested, though some are more plausible than others and that is true for the many additional alternatives one could think of. We do however feel that the observed differences in connectivity, as nicely formulated by the reviewer under point 1 allow for some speculative comments of why this might be relevant. We have rephrased the relevant parts of the discussion accordingly.

The reviewer is apparently not satisfied with the additional supplementary data and the arguments we provided to address the potential risk of selectivity in the population of neurons in the particular TG animal. Of course, we cannot exclude that option but in the previous revision we provided arguments to address this. This was embedded in the result section and we feel that it would interrupt the flow of the manuscript to move that into the discussion. Our arguments are on page 8, lines 202-211. We have reiterated this now in the discussion in two sentences on page 18, lines 457460 which reads: ‘Note that in the present study we focussed on the presumed direct excitatory connectivity from LVb-to-LVa neurons using a newly derived TG mouse line. Although we above provided data to argue that we find it unlikely that the reported difference in connectivity between LEC and MEC might be caused by a different preference for a certain cell type in MEC versus LEC, we cannot completely exclude that option.

3. The possibility that differences in cut dendrites explain the differences in synaptic responses between LEC and MEC, and within the MEC, is still not ruled out. The revised manuscript now gives an indication of the number of dendritic branches in MEC that are > 150 μm. This is insufficient as layer 3 can be on the order of 300 μm or more in width between its superficial and deep borders, while the axons from L5b extend well into L2. To make the claims convincing it's important that adequate quantification of remaining apical dendritic length (and if possible diameter/surface areas) in LEC and at each dorsoventral level of MEC is included (for both cut and not-cut neurons).

We have now provided supplementary data (Figure 4—figure supplement 2D) providing detailed information on how far dendrites extend towards superficial layers for each neuron and correlated that parameter with the evoked postsynaptic responses. The data indicate that it is unlikely that the severing of the apical dendrites caused the differences in the response between MEC- and LEC-LVa neurons. Our material does not allow to provide information of dendritic dimensions other than distribution across layers, so we have therefor not included diameter and surface area information.

We further did not include potential differences in receptor/channel distributions, all of which might result in differences not only between neurons with or without cut dendrites, but also between MEC and LEC neurons with intact dendrites. We thus suggest, and agree with the reviewer, that a parsimonious explanation is the striking difference in apparent anatomical density of the projections from LVb to LVa.

4. I couldn't find anywhere whether any connections were tested with glutamate receptor antagonists to confirm they are synaptic and glutamatergic. It's unlikely but not impossible that some short latency responses reflect low level expression of ChR2 in non-5b neurons (in my experience this might not be detectable from fluorescence). It would be good to know this can be ruled out.

We did not use any pharmacological approaches, as suggested by the reviewer, to test whether contacts are synaptic and glutamatergic. Note that we analyzed the transgenic line extensively and we report that the line almost exclusively labels non-GABAergic neurons: ‘using a GAD67 transgenic line expressing green fluorescent protein (GFP), we showed that the percentage of double-labelled (PCP4+, GAD67+) neurons among total GAD67-positive neurons is very low in both LEC and MEC (4.3% and 2.3% respectively, Figure 1—figure supplement 2)’.

Therefore, we assume that most of the TG-targeted neurons are excitatory, likely glutamatergic. This is certainly something we will explore in the future as we plan to do an ‘in vivo’ study using the same mouse line.

We cannot exclude that a number of tetracycline-controlled transactivator (tTA, Tet-Off) neurons expressed too weakly to be reported in the tTA-dependent reporter mouse which expresses mCherry and that such non-reported neurons might express ChR2 at a level that would impact our results, particularly through representing an unknown class of non-glutamatergic neurons, but we deem that rather unlikely. Since we were looking for overall connectivity differences or similarities between LEC and MEC, and in view of the striking differences between responses in LVa in dorsal LEC versus dorsal MEC we did not explore either of these possibilities further.